# Evolutionary adaptation of bacterial proteomes to translation-impeding sequences

Keigo Fujiwara [1,2,3,4]✉, Naoko Tsuji[1,4], Karen Sakiyama[1,4], Hironori Niki[3] & Shinobu Chiba [1,2]✉

## Abstract

Microbial translation arrest peptides monitor intracellular environments and feedback-regulate downstream gene expression. Previous studies have identified a class of bacterial arrest peptides with C-terminal RAPP-like sequences, encoded upstream of genes involved in protein localization. In this study, we found that among RAPP-like sequences, RAPP (Arg-Ala-Pro-Pro) and RGPP (Arg-Gly-Pro-Pro) could more readily evolve into translation-impeding sequences with a particularly robust arrest that is refractory to EF-P. RAPP-like motifs were found to be strongly excluded from bacterial proteomes, likely reflecting the risk of disrupting the cellular translation system. Meanwhile, these motifs tended to occur near the C-terminus of relatively small secretory and membrane proteins. Notably, they were encoded upstream of genes with diverse functions beyond protein localization. Indeed, we identified seven RAPP/RGPP-containing arrest peptides from *Streptomyces lividans* encoded upstream of genes with diverse functions. These findings illustrate the bidirectional evolution of RAPP-containing proteins: their elimination from bacterial proteomes and their adaptation into arrest peptides with various regulatory roles.

**Keywords** Arrest Peptide; Ribosome; SecM; Translation Arrest
**Subject Categories** Evolution & Ecology; Microbiology, Virology & Host Pathogen Interaction

## Introduction

The arrest peptide co-translationally stalls the ribosome through interacting with ribosomal components near the peptidyl transferase center (PTC) and the nascent polypeptide exit tunnel (NPET) (Chiba et al, 2023; Dever et al, 2020; Ito and Chiba, 2013; Wilson et al, 2016). Translation arrest is typically responsive to changes in the intracellular environments, enabling arrest peptides to serve as sensors in the feedback regulation of downstream gene expression (Ito et al, 2018). Cofactor-dependent arrest peptides induce translation arrest in response to specific metabolites or antibiotics, thereby acting as sensors for corresponding cofactors (Bhattacharya

et al, 2024; Vázquez-Laslop and Mankin, 2018). An intrinsic class of arrest peptides, exemplified by *Escherichia coli* SecM (Nakatogawa and Ito, 2001), *Bacillus subtilis* MifM (Chiba et al, 2009), *Vibrio alginolyticus* VemP (Ishii et al, 2015) induces translation arrest in a cofactor-independent manner. The arrest caused by these arrest peptides is released when they are pulled from the N-terminus during translocation across or into the membrane (Chiba et al, 2011; Nakatogawa and Ito, 2001). Prolonged arrest due to deficiencies in the Sec- or YidC-dependent localization pathways leads to the induction of downstream genes encoding Sec components or YidC, ensuring the maintenance of homeostasis of these pathways (Chiba et al, 2009; Ishii et al, 2015; Chiba and Ito, 2015; Nakatogawa and Ito, 2001). Despite their functional similarity, these arrest peptides share no sequence similarity to one another (Nakatogawa and Ito, 2002; Chiba et al, 2009; Fujiwara et al, 2018; Chiba and Ito, 2012; Ishii et al, 2015; Mori et al, 2018). For example, SecM arrest sequence is characterized as FxxxxWIxxxxGIRAGP (Nakatogawa and Ito, 2002), whereas MifM arrest sequence is characterized as RIxxWIxxxxxMNxxxxDEED (Chiba et al, 2009; Chiba and Ito, 2012; Sohmen et al, 2015) (x indicates non-essential residues).

Recently, we identified nearly 20 arrest peptides encoded upstream of genes involved in the protein localization pathways (Fujiwara et al, 2024; Sakiyama et al, 2021). Unexpectedly, many of the newly identified arrest peptides were found to share a RAPP-like common sequence on their C-termini. For example, ApcA from *Rhodococcus erythropolis* carries the RAPG sequence, whereas *Amycolatopsis japonica* ApdA and *Sinorhizobium medicae* ApdP carry the RAPP sequence (Sakiyama et al, 2021). Some homologs of ApdA and ApdP carry the RGPP sequence (Fujiwara et al, 2024; Sakiyama et al, 2021). The C-terminal RAP sequence of ApdA and ApdP adopts an identical conformation within the stalled ribosome, in which the peptidyl transfer from the P-site peptidyl-tRNA to the A-site amino acyl-tRNA was abolished (Morici et al, 2024). Remarkably, SecM, whose arrest sequence ends with the RAGP sequence, was also shown to share the identical conformation near the PTC (Gersteuer et al, 2024), suggesting the existence of a common underlying mechanism.

Our previous study of ApcA, ApdA, and ApdP has shown that their arrest-essential sequences consist of both the RAPP-like C-terminal core region, and the N-terminal adjacent region, the latter of which bears no sequence similarity to each other (Sakiyama et al, 2021). Despite sharing a similar sequence in the C-terminal core

[1]Faculty of Life Sciences, Kyoto Sangyo University, Kyoto, Japan. [2]Institute for Protein Dynamics, Kyoto Sangyo University, Kyoto, Japan. [3]Department of Gene Function and Phenomics, National Institute of Genetics, Mishima, Japan. [4]These authors contributed equally: Keigo Fujiwara, Naoko Tsuji, Karen Sakiyama.✉E-mail: kig.fujiwara@nig.ac.jp; schiba@cc.kyoto-su.ac.jp

region, their arrest appears to require different interactions through the N-terminal region (Sakiyama et al, 2021; Morici et al, 2024). However, it remains unclear what specific interactions in the N-terminal region support the arrest-inducing conformation of the C-terminal core region, and which specific sequences or residues in the C-terminal core region influence the N-terminal interactions. Furthermore, the functional diversity of RAPP-containing arrest peptides was also not well understood.

In this study, we began with a comprehensive mutagenesis of the C-terminal core regions of ApcA, ApdA, and ApdP to determine their crucial properties. The results reveal the importance of specific amino acid residues near the PTC. In contrast, the N-terminal adjacent region exhibited greater flexibility, particularly with RAPP and RGPP sequences, which support arrest in combination with N-terminal sequences from different arrest peptides. This observation suggests that RAPP-like sequences, particularly RAPP and RGPP, have the potential to readily evolve into translation-stalling sequences. Our bioinformatics analysis reveals that RAPP-like sequences are strongly avoided in bacterial proteomes, possibly to mitigate the risk of the occurrence of translation-impeding sequences. On the other hand, RAPP and RGPP sequences are frequently located near the C-terminus of relatively small proteins, suggesting that many bacteria employ these sequences to evolve arrest peptides that regulate the expression of their downstream genes. Remarkably, the proteins encoded by their downstream genes include those with diverse functions beyond protein localization pathways. Indeed, we identified seven RAPP/RGPP-containing arrest peptides from *Streptomyces lividans*, which are encoded upstream of genes with diverse functions. These findings support the possibility that RAPP-containing arrest peptides are widespread and involved in various biological processes.

## Results

### Identification of crucial features of the RAP sequence of ApcA, ApdA, and ApdP for ribosome stalling

ApcA, ApdA, and ApdP share similar C-terminal sequences (RAPG in ApcA, and RAPP in ApdA and ApdP). The ribosome stalling occurs when the third and fourth codons of the RAPG/P sequence occupy the P-site and A-site, respectively (Sakiyama et al, 2021). To identify crucial features of the C-terminal core region, we conducted systematic mutagenesis targeting this motif. Particularly, we focused on the first three residues, R-A-P, because our previous study targeted the fourth (A-site) codon (Sakiyama et al, 2021). Ribosome stalling was assessed using in vivo LacZ reporter assay. Gene segments encoding the C-terminal soluble domains of these arrest peptides were fused in-frame between those encoding GFP and LacZ (GFP-'AP-lacZ; Fig. 1A). Then, each of the RAP motifs was systematically mutated to all other 19 amino acid residues.

*B. subtilis* strains expressing the wild-type GFP-ApcA-LacZ exhibited low β-galactosidase activity (14.3 units; R in Fig. 1B; R102) due to translation arrest upstream of *lacZ*. In contrast, all 19 Arg-substituted mutant derivatives exhibited high levels of β-galactosidase activity exceeding 60 units (Fig. 1B; R102). Similar results were obtained for the systematic mutagenesis of Ala103 and

Pro104 (Fig. 1B; A103, A104), demonstrating that only the wild-type R-A-P sequence allows efficient ribosome stalling of ApcA.

Similarly, all mutations of Arg120 or Pro122 of ApdA resulted in high levels of β-galactosidase activity (>20 units, Fig. 1C; A120, P122). In contrast, the substitution of Ala121 with Gly led to lower β-galactosidase activity (2.4 units) compared to the wild-type (3.0 units, Fig. 1C; A121). Substitutions of Ala121 to Ser, Asp, and Pro resulted in intermediate levels of β-galactosidase activity (9.0, 10.0, and 12.9 units, respectively). Other mutants exhibited high levels of β-galactosidase activity. These results suggest that the A121G mutant of ApdA is fully permissive or even enhances translation arrest, whereas the A121S, A121D, and A121P mutants were partially permissive.

All of the *B. subtilis* strains expressing ApdP mutant derivatives with Arg131 substituted by other amino acids elevated β-galactosidase activity (Fig. 1D; R131). This was the case when they were expressed in *E. coli* (Fig. 1E; R131). Interestingly, the *B. subtilis* reporter strain expressing the A132G mutant derivative exhibited lower β-galactosidase activity (0.9 units) compared to the wild-type (1.9 units) (Fig. 1D; A132). The *B. subtilis* A132P reporter strain also showed low β-galactosidase activity (2.4 units) (Fig. 1D; A132). The *B. subtilis* reporters with the A132D, A132V, A132S, and A132E substitutions exhibited intermediate levels of β-galactosidase activity (11.3, 14.8, 15.9, and 19.1 units, respectively) (Fig. 1D; A132). Similar results were obtained from the reporter assay using *E. coli*. The A132G substitution resulted in much lower β-galactosidase activity (21.4 units) compared to the wild-type (151.8 units) (Fig. 1E; A132), while the A132P, A132D, A132S, and A132E substitutions resulted in intermediate levels of β-galactosidase activity (491.0, 696.9, 1088.3, and 1548.8 units, respectively). These results suggest that, regardless of whether *E. coli* or *B. subtilis* strains were used, the A132G mutation increased the translation arrest activity of ApdP, while the A132P, A132D, A132S, and A132E substitution partially support the arrest-inducing ability, revealing a remarkable similarity to the observations obtained from the mutagenesis of Ala121 in ApdA.

The substitution of Pro133 in the ApdP reporter elevated β-galactosidase activity in almost all cases, except for P133G. This mutant exhibited significantly lower β-galactosidase activity compared to other mutants, regardless of whether it was expressed in *B. subtilis* or *E. coli* (6.4 units in *B. subtilis* and 333.1 units in *E. coli*) (Fig. 1D,E; P133). These results suggest that the substitution of Pro133 with Gly partially supports the arrest-inducing ability of ApdP. Notably, the resulting RAGP sequence is identical to the C-terminal sequence of the SecM arrest motif ($F_{150}$XXXXWIXXXXGIRAGP$_{166}$) (Nakatogawa and Ito, 2002).

In summary, our systematic mutagenesis demonstrates that (i) The R-A-P sequence of these arrest peptides exhibits high sequence specificity, allowing only limited substitutions. In particular, the Arg in the RAP sequence is irreplaceable by other amino acids; (ii) Substitution of Ala in the RAP sequence of ApdA, and ApdP with Gly is fully permissive or even enhances arrest activity. This is consistent with the observation that the RGPP sequence is sometimes utilized in homologs of ApdA and ApdP (Fujiwara et al, 2024; Sakiyama et al, 2021). (iii) Substitutions of Ala with other relatively small amino acids, such as Ser, Asp, or Pro, partially retain the arrest-inducing activity of ApdA and ApdP. Buskirk and colleagues also identified RxPP sequences (where x is Gly, Ala, Ser, or Pro) as part of arrest-inducing sequences through their screening

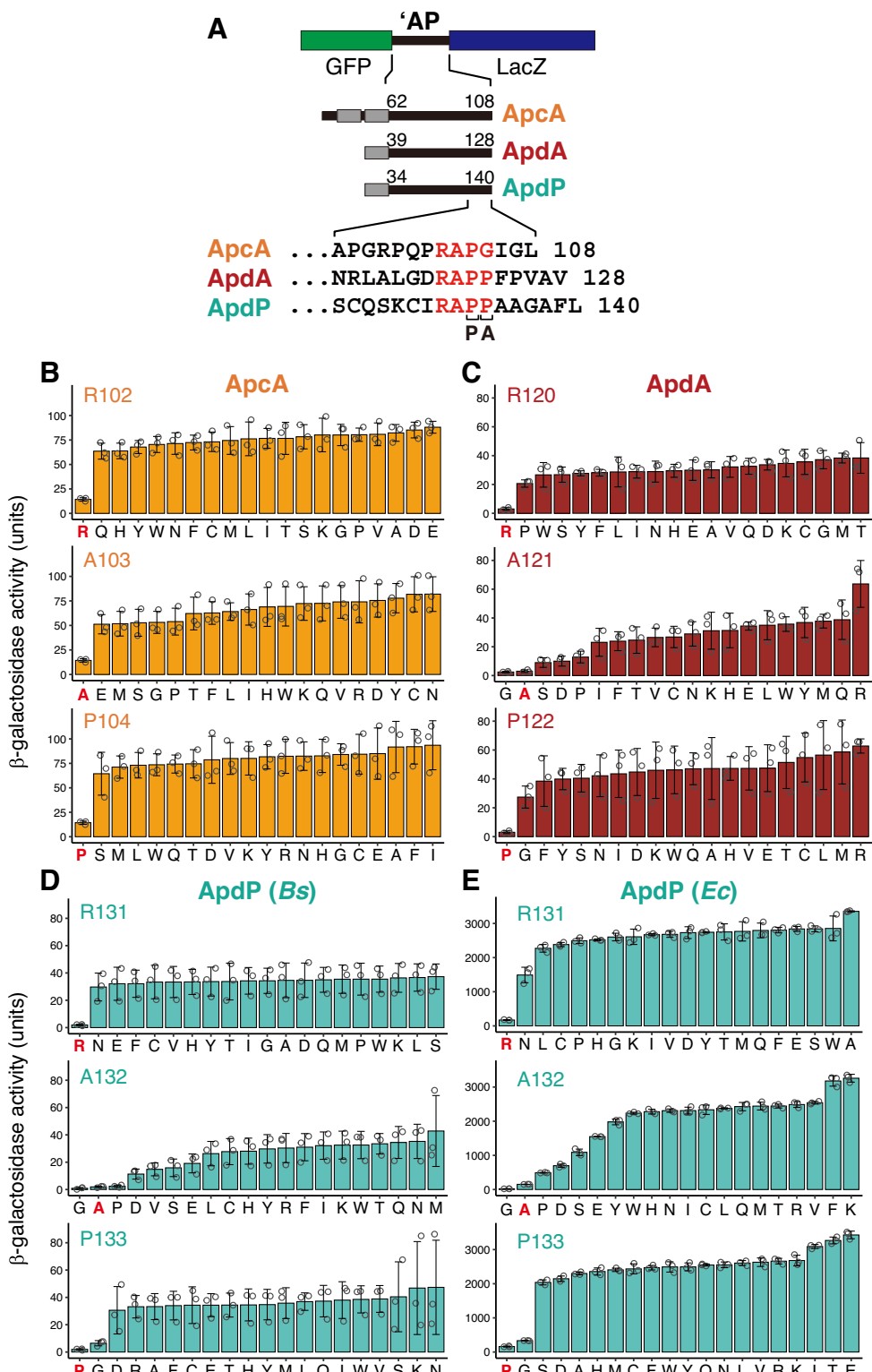

**Figure 1. Systematic mutagenesis of the RAP motif of ApcA, ApdA, and ApdP.**

(A) Schematic representation of the *lacZ*-based arrest reporter (GFP-'AP-lacZ) used for the in vivo LacZ reporter assay. The amino acid numbers and sequences of ApcA, ApdA, and ApdP are shown. 'AP' indicates the C-terminal moiety of the arrest peptide, while 'P' and 'A' at the bottom indicate P-site and A-site residues, respectively. Gray boxes represent transmembrane segments (ApcA) or signal sequences (ApdA and ApdP). (B–E) The β-galactosidase activity (mean, $n = 3$ biologically independent cell cultures) of the reporters for ApcA (B), ApdA (C), and ApdP (D, E). *B. subtilis* (B–D) or *E. coli* cells (E) expressing wild-type or mutant derivatives of the *lacZ*-based reporters were cultured at 37 °C and subjected to the β-galactosidase assay. The residues targeted by mutation are indicated in the upper left corner, and the residues introduced are indicated at the bottom of each graph. Wild-type reporters are labeled with the original residue name in bold red (i.e., R in R102 of ApcA). The error bars and dots represent standard deviations and individual data points, respectively. Source data are available online for this figure.

using a random sequence library in *E. coli* (Tanner et al, 2009; Woolstenhulme et al, 2013); (iv) The RAGP sequence can support the arrest of ApdP, highlighting a similarity between ApdP and SecM. The irreplaceability of Arg was also previously demonstrated for SecM (Yap and Bernstein, 2009).

## Identification of versatile C-terminal core sequences that support arrest in diverse contexts

Previous analysis showed that the sequence RAPP supports translation arrest in the contexts of ApcA (Sakiyama et al, 2021) and SecM (Tanner et al, 2009; Gersteuer et al, 2024), as well as in the RAPP-containing ApdA and ApdP. To further explore the existence of other versatile core motifs, we examined the arrest-supporting ability of the RGPP, RAPG, and RAGP sequences in the context of different arrest peptides.

The substitution of the RAPP sequence in ApdP with RAPG elevated β-galactosidase activity to a level similar to that of AAPP or AAGP mutants in both *B. subtilis* and *E. coli* (Fig. 2A,B), suggesting that the RAPG sequence cannot support translation arrest in the context of ApdP. In contrast, the RAPP version of GFP-ApcA-lacZ exhibited significantly lower β-galactosidase activity than wild-type (1.2 units vs 13.3 unit; Fig. 2C), as reported previously (Sakiyama et al, 2021). The RGPP versions exhibited an even lower level of β-galactosidase activity (0.9 units), suggesting that RAPP and RGPP sequences enhanced ribosome stalling in the context of ApcA. Interestingly, the reporter strain expressing the RAGP (SecM-type) sequence also showed reduced β-galactosidase activity (7.7 units; Fig. 2C).

The *E. coli* strain expressing the GFP-SecM(38-170)-LacZ reporter exhibited low β-galactosidase activity (2.8 units). As reported previously (Gersteuer et al, 2024), a mutant derivative with the RAPP instead of the $R_{163}AGP_{166}$ sequence showed low β-galactosidase activity (2.1 units; Fig. 2D). The RGPP version exhibited even lower β-galactosidase activity (0.6 units; Fig. 2D). In contrast, the RAPG version exhibited high β-galactosidase activity, similar to known arrest-defective R163A (AAGP) and P166A (RAGA) variants (Nakatogawa and Ito, 2002), as well as the AAPP variant. These results suggest that RAPP and RGPP sequences function as versatile arrest-supporting motifs in the contexts of ApcA, ApdA, ApdP, and SecM. Particularly, the RGPP sequence generally supports the most robust translation arrest.

## EF-P does not release ribosome stalling caused by RAPP-containing arrest peptides

The stable in vivo arrest of RAPP-containing arrest peptides suggests that their arrest may be refractory to EF-P, which generally facilitates the synthesis of consecutive proline sequences (Peil et al, 2013; Ude et al, 2013; Doerfel et al, 2013). To test this possibility, we examined the effect of EF-P on the arrest of ApdA and ApdP using original and modified PURE systems. The original PURE system (hereafter referred to as Ec PURE) consists of purified *E. coli* ribosome, translation components derived from *E. coli*, and T7 RNA polymerase (Shimizu et al, 2001). The modified system, referred to as Bs hybrid PURE system (or Bs PURE in short) was developed by replacing the *E. coli* ribosomes in Ec PURE with *B. subtilis* ribosomes (Chiba et al, 2011).

DNA fragments encoding reporter proteins composed of GFP, C-terminal soluble domains of the arrest peptides, and LacZα (Fig. 2E) were used as templates for in vitro transcription–translation reaction. Translation arrest products have C-terminal tRNA (Fig. 2E; peptidyl-tRNA), thus exhibiting slower migration on SDS–PAGE compared to the RNase-pretreated arrest products (Fig. 2E; arrest product). Compatibility of *E. coli* EF-P with Bs PURE was confirmed using a reporter containing five consecutive prolines or alanines between GFP and LacZα (referred to as 5P and 5A, respectively, Fig. EV1A,B, lanes 1–6; upper and lower panels).

We then examined the EF-P sensitivity of ApdA and ApdP. Translation of the ApdA reporter in Bs PURE accumulated the arrest product almost exclusively, regardless of the presence or absence of EF-P (Fig. EV1B, lanes 7–10; lower panel). Similarly, translation of ApdP in either the Ec PURE or Bs PURE resulted in the accumulation of the arrest product, and it was not affected by the addition of EF-P (lanes 13–16). These results demonstrate that the translation arrest of ApdA and ApdP are not rescued by EF-P.

## RAPP and RGPP sequences support translation arrest with different N-terminal sequences in vitro

To further confirm that the RAPP and RGPP sequences support or enhance arrest in the contexts of ApdP, ApcA, and SecM, we employed Ec and Bs PURE systems with EF-P. Translation of the wild-type (RAPP), RGPP, and RAGP mutant derivatives of the ApdP reporter in Bs PURE resulted in a predominant accumulation of arrest products (Fig. 2F, lanes 1–6). In contrast, the RAPG and AAPP mutants predominantly accumulated as the full-length products (Fig. 2F, lanes 7–10). These results are in good agreement with the in vivo reporter assay (Fig. 2A). Similarly, translation of ApdP reporters in Ec PURE showed results consistent with the in vivo reporter assay (Fig. 2B), except for the RAGP mutant derivative, which predominantly accumulated as the full-length product in vitro (Fig. 2G).

Translation of the RAPP and RGPP versions of the ApcA reporter in Bs PURE resulted in increased accumulation of the arrest products with a concomitant reduction in full-length products (Fig. 2H, lanes 1–4) compared to the wild-type (RAPG) and RAGP versions (lanes 5–8). These results are consistent with in vivo results, suggesting that RAPP and RGPP sequences enhance arrest, and the RAGP sequence supports it. The arrest-defective AAPP derivative serves as a control (lanes 9, 10).

Translation of wild-type GFP-SecM-LacZα (RAGP) accumulated the major arrest products and minor full-length products (Fig. 2I). In contrast, the RAPP and RGPP variants accumulated exclusively as the arrest products (Fig. 2I), suggesting that substituting RAGP with RAPP or RGPP enhances arrest. The RAPG and AAPP derivatives accumulated as the full-length species (Fig. 2I). These results are consistent with the in vivo assay (Fig. 2D). In summary, our in vivo and in vitro studies demonstrate that the RAPP and RGPP sequences support translation arrest in the contexts of ApcA, ApdA, ApdP, and SecM.

## The C-terminal core sequence affects the N-terminal interactions with the ribosome

The observation that RAPP and RGPP sequences can be combined with a wider variety of sequences to support robust translation arrest suggests that

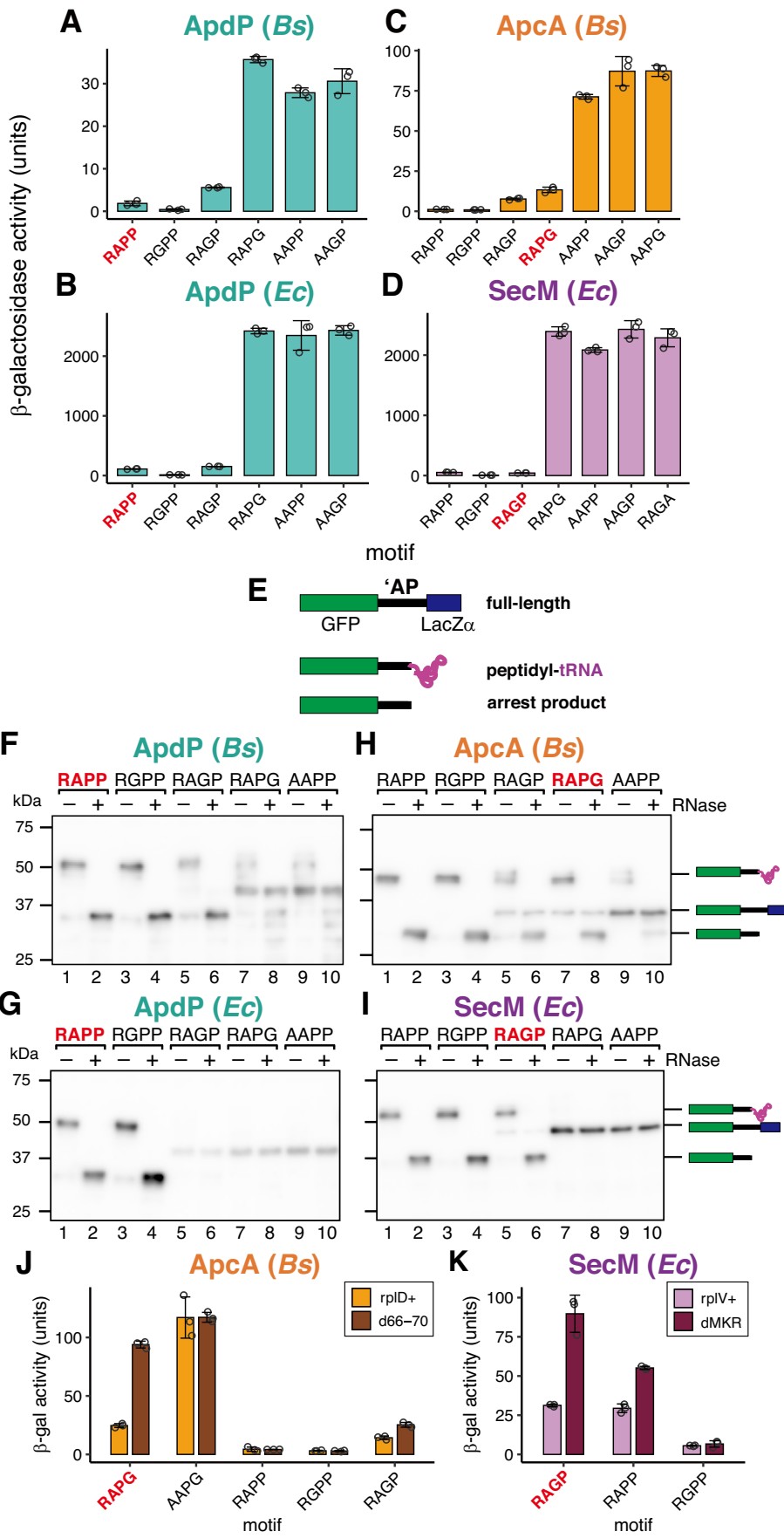

◀ **Figure 2. RAPP and RGPP support translation arrest with different N-terminal sequences.**

(A–D) The β-galactosidase activity (mean, n = 3 biologically independent cell cultures) of the reporters for ApdP (**A, B**), ApcA (**C**), and SecM (**D**). *B. subtilis* (**A, C**) or *E. coli* cells (**B, D**) expressing *lacZ*-based reporters were cultured at 37 °C and subjected to the β-galactosidase assay. The sequence of the C-terminal motif of each reporter is indicated at the bottom. Wild-type reporters are indicated with their original sequence (in bold red). (**E**) Schematic representation of the reporter used for the in vitro translation arrest assay. The tRNA is depicted in magenta. (**F–I**) Western blot analysis of the in vitro translation products. Reporter genes harboring wild-type (red bold) or mutant (black) arrest motifs, as indicated at the top, were translated in the *Bs* (**F, H**) or *Ec* (**G, I**) PURE systems. Translation products were separated using neutral-pH gels and immunoblotted with anti-GFP antibody. Prior to the separation, a portion of the samples was treated with RNase A (lanes indicated as +) to degrade the tRNA moiety. Molecular size standards are indicated by horizontal lines on the left of each membrane. (**J, K**) β-galactosidase activity (mean, n = 3 biologically independent cell cultures) of the reporters for ApcA (**J**), and SecM (**K**) using *B. subtilis* (**J**) or *E. coli* cells (**K**). The sequence of the C-terminal motif of each reporter is indicated at the bottom. The error bars and dots represent standard deviations and individual data points, respectively. Source data are available online for this figure.

that RAPP/RGPP-containing arrest peptides may have relatively lower requirements for N-terminal interactions to induce arrest. Indeed, ApdA and ApdP can induce arrest with relatively short N-terminal regions and are unaffected by mutations in uL4 and uL22 at the constriction sites of the NPET (Morici et al, 2024). In contrast, the arrest sequences of ApcA and SecM are relatively long, and translation arrest of SecM is weakened by mutations in uL22 (Nakatogawa and Ito, 2002; Morici et al, 2024). If this assumption is correct, the translation arrest of the RAPP or RGPP versions of ApcA or SecM might be less affected by ribosome mutations compared to their wild-type counterparts. To test this possibility, we investigated the effect of tunnel mutations on the translation arrest of RAPP and RGPP versions of ApcA and SecM.

The *rplD* mutant strain (*d66-70*), which has a deletion of five residues (66–70), exhibited higher β-galactosidase activity when expressing the wild-type ApcA reporter compared to the wild-type *rplD⁺* strain (Fig. 2J, RAPG), suggesting that the translation arrest of ApcA is compromised by the uL4 mutation. The AAPG version, which lacks translation arrest capability, showed high levels of β-galactosidase activity in both the wild-type and mutant strains. Interestingly, the RAPP and RGPP versions exhibited low levels of β-galactosidase activity in both *rplD⁺* and *d66-70* strains (Fig. 2J). The *d66-70* mutant strain expressing the RAGP version showed slightly higher activity than the *rplD⁺* strain. These results suggest that the presence of RAPP or RGPP sequences enables ApcA to induce strong arrest without interacting with uL4.

We conducted a similar analysis for SecM, whose arrest is compromised by uL22 mutations (Nakatogawa and Ito, 2002). We overexpressed either a wild-type (*rplV⁺*) or a mutant (*rplV-dMKR*) derivative of uL22 in *E. coli*, which has a wild-type copy of *rplV+* on the chromosome. The *rplV-dMKR* has a deletion of residues $M_{82}KR_{84}$ of uL22 in the NPET. The *dMKR* strain expressing the wild-type SecM version (RAGP) exhibited β-galactosidase activity of ~2.9-fold higher than the wild-type strain, confirming that the arrest of SecM is compromised by the uL22 mutation. The RAPP version of SecM also showed higher β-galactosidase activity in the *dMKR* strain compared to the wild-type strain, albeit with a lower extent (1.9-fold, Fig. 2K; RAPP) compared to the wild-type RAGP version. When expressing the RGPP version, both wild-type and *dMKR* strains showed similarly low β-galactosidase activity (only 1.2-fold difference). These results suggest that RAPP and RGPP sequences enable the creation of arrest sequences that do not rely on interactions with the constriction site.

## Bacterial proteomes generally avoid translation-impeding sequences

Stalling sequences could potentially be disadvantageous for general protein synthesis and would, therefore, be underrepresented in the proteome. To investigate this hypothesis, we analyzed the occurrence frequency of four consecutive-amino-acid sequences in bacterial proteomes using an in silico approach. We calculated the "frequency bias" of each four-amino-acid sequence by counting its actual frequency in bacterial proteomes and normalized it by the expected frequency, which accounted for the frequency of each amino acid (Appendix Fig. S1). For example, *E. coli* MG1655 has no RGPP-containing proteins and one RAPP-containing protein (YhfW). This results in frequency biases of 0 and 0.073, respectively. The RAGP motif was found in five proteins (SecM, EutC, GlpG, IlvD, YigB), resulting in a frequency bias of 0.22. The *B. subtilis* 168 proteome contains no RAPP or RGPP-containing proteins but one RAGP-containing protein (YwcI), which caused translation arrest (Fujiwara et al, 2024). Thus, the frequency bias of RAGP in *B. subtilis* is 0.103.

To assess the distribution of frequency bias across bacterial species, we calculated the frequency bias for each bacterial species in a representative dataset of one thousand bacterial species from major phyla, and generated the empirical cumulative distribution function (ECDF) plots (Fig. 3A). Among the 1000 proteomes in the γ-proteobacteria class, 40.9% completely lack RGPP sequences. Frequency bias values for almost all of them (99.9%) are below 1, indicating that RGPP sequence is underrepresented in most of the species in the γ-proteobacteria class (Fig. 3A, red line). RAPP and RAGP are found in 88.4% and 91.9% of the γ-proteobacteria proteomes, respectively; however, their frequency biases are lower than 1 in most bacterial species (Fig. 3A, blue and green lines). Similar results were obtained for α-proteobacteria (Fig. 3A), where species with a frequency bias exceeding 1 accounted for 1% or less, Bacillota, Actinomycetota, the major groups of gram-positive bacteria (Fig. 3A), and others (Fig. EV2). Specifically, 99.3% of Bacillota proteomes lack RGPP and/or RAPP, while more than 70.9% lack RAGP.

The mean frequency bias of four-amino-acid sequences was calculated for each of the 17 major bacterial phyla or classes, and the results for those with notably low values were visualized using a heatmap (Figs. 3B and EV3A). Remarkably, the mean frequency of the RGPP sequence was the lowest for the 10 major bacterial groups, indicating that it is the most underrepresented sequence among 160,000 four-amino-acid sequence patterns in these phyla. Similarly, the mean frequency of RAPP was the lowest for 2 phyla. RDPP, RSPP, and RAGP were also highly underrepresented in diverse bacterial proteomes. As aforementioned, our mutational analysis of ApdA and ApdP showed that substitutions of Ala to Asp or Ser, yielding RDPP or RSPP, were partially permissive, and RAGP is the motif in the SecM arrest sequence (Fig. 1C–E). These results suggest that motifs with the potential to evolve into

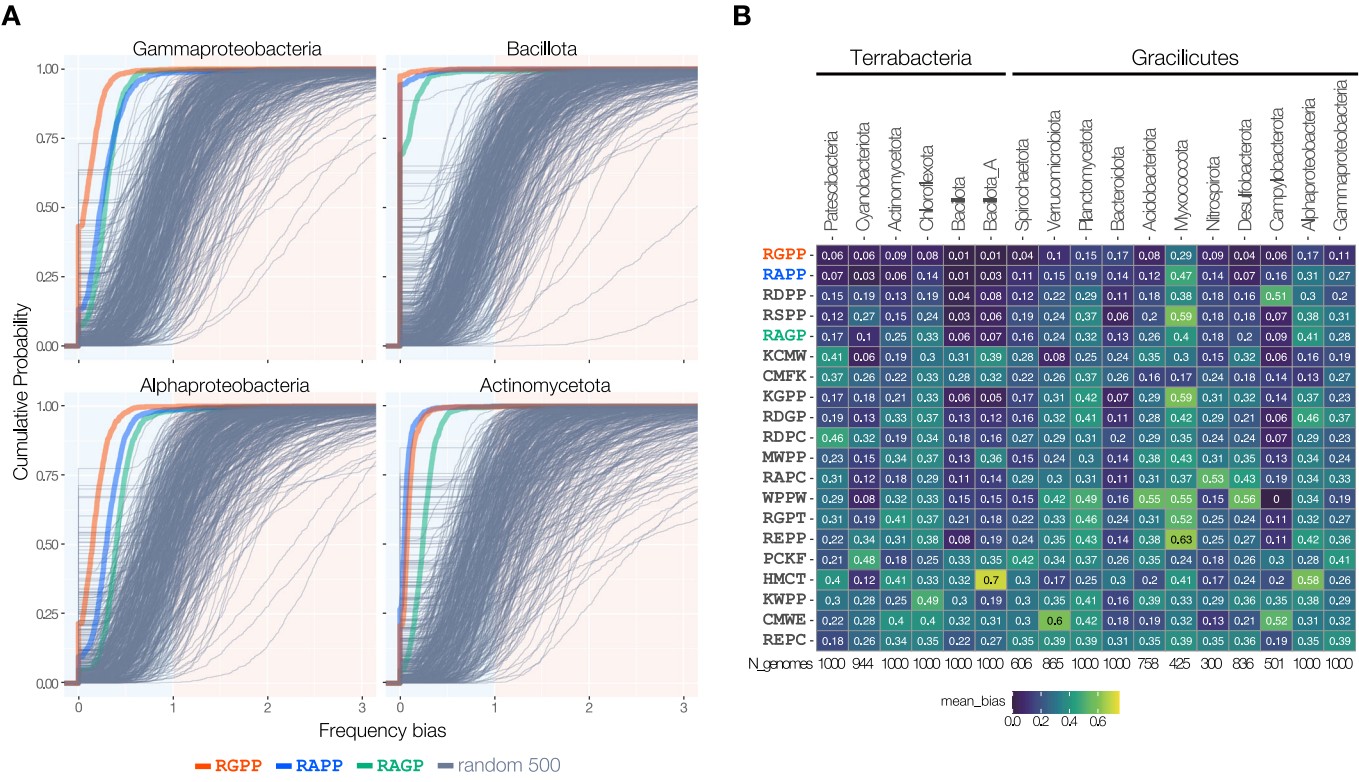

**Figure 3. RAPP-like sequences are generally avoided in bacterial proteomes.**

(A) Empirical cumulative distribution functions (ECDFs) of frequency bias for four-amino-acid sequences in Gammaproteobacteria, Alphaproteobacteria, Bacillota, and Actinomycetota. Frequency bias values for the RGPP (red), RAPP (blue), RAGP (green), and randomly selected 500 patterns are plotted. (B) Heatmap showing the average frequency bias of four-amino-acid sequences with low frequency bias. The average frequency bias for each four-amino-acid sequence was calculated across 15 phyla and two major classes of the phylum Pseudomonadota. The 20 patterns with the lowest values across bacteria are shown. A heatmap including top 50 patterns is provided in Fig. EV3A. Source data are available online for this figure.

translation-impeding sequences have been significantly eliminated from bacterial proteomes over the course of evolution.

## RAPP/RGPP-containing small proteins share common features with arrest peptides

To determine whether proteins containing RAPP-like motifs share common structural or functional features, we conducted further in silico analyses. We extracted protein sequences containing RGPP or RAPP from the predicted proteomes of 52,895 Genome Taxonomy Database (GTDB) representative genomes (Parks et al, 2018). Among a total of 168,430,932 proteins, RGPP and RAPP motifs were found in 88,578 and 186,182 proteins, respectively (Fig. EV4A). These numbers were significantly lower than those containing AGPP and AAPP motifs (686,857 and 1,807,575). A comparison of protein lengths between those containing RAPP or RGPP sequences and those with AAPP or AGPP sequences revealed that proteins with RAPP or RGPP sequences tend to be smaller in size (Fig. EV4B). Notably, the ECDF plot for proteins with RAPP or RGPP sequences exhibited a characteristic shoulder around the 100–200 amino acid range (Fig. EV4B). Next, we focused on small proteins with fewer than 300 amino acid residues and plotted the positions of each motif relative to its distance from the C-terminus (Fig. EV4C). The analysis revealed that RAPP and RGPP motifs tended to occur closer to the

C-terminus compared to AAPP and AGPP motifs (Fig. EV4C). In addition, compared to AAPP and AGPP motifs, RAPP and RGPP motifs were more commonly found in secretory proteins, with this tendency being particularly pronounced in small proteins with fewer than 300 residues (Fig. EV4D).

These features are reminiscent of the sec/yidC-associated arrest peptides, which regulate the downstream gene expression (Chiba et al, 2009; Ishii et al, 2015; Nakatogawa and Ito, 2001; Fujiwara et al, 2024). Thus, we focused on the downstream genes of RAPP and RGPP-containing small ORFs (<300 residues). Information on genes located downstream of those encoding RAPP-, RGPP-, AAPP-, and AGPP-containing proteins was collected and classified based on their annotated functions. We then determine the ratios of uORFs encoding the RAPP motif relative to those encoding the AAPP motif, as well as those encoding the RGPP motif relative to those encoding the AGPP motif, to identify groups of genes with high RGPP/AGPP (Fig. 4A; Appendix Fig. S2A) and RAPP/AAPP ratios (Fig. 4B; Appendix Fig. S2B).

As expected, annotation classes with high RGPP/AGPP or RAPP/AAPP ratios included sec genes such as secA, and secD (Fig. 4A,B; outlined in orange rectangle), indicating that genes encoding RAPP/RGPP motifs are overrepresented upstream of sec genes. However, we also identified many downstream genes with different functions. For example, genes encoding TonB-dependent

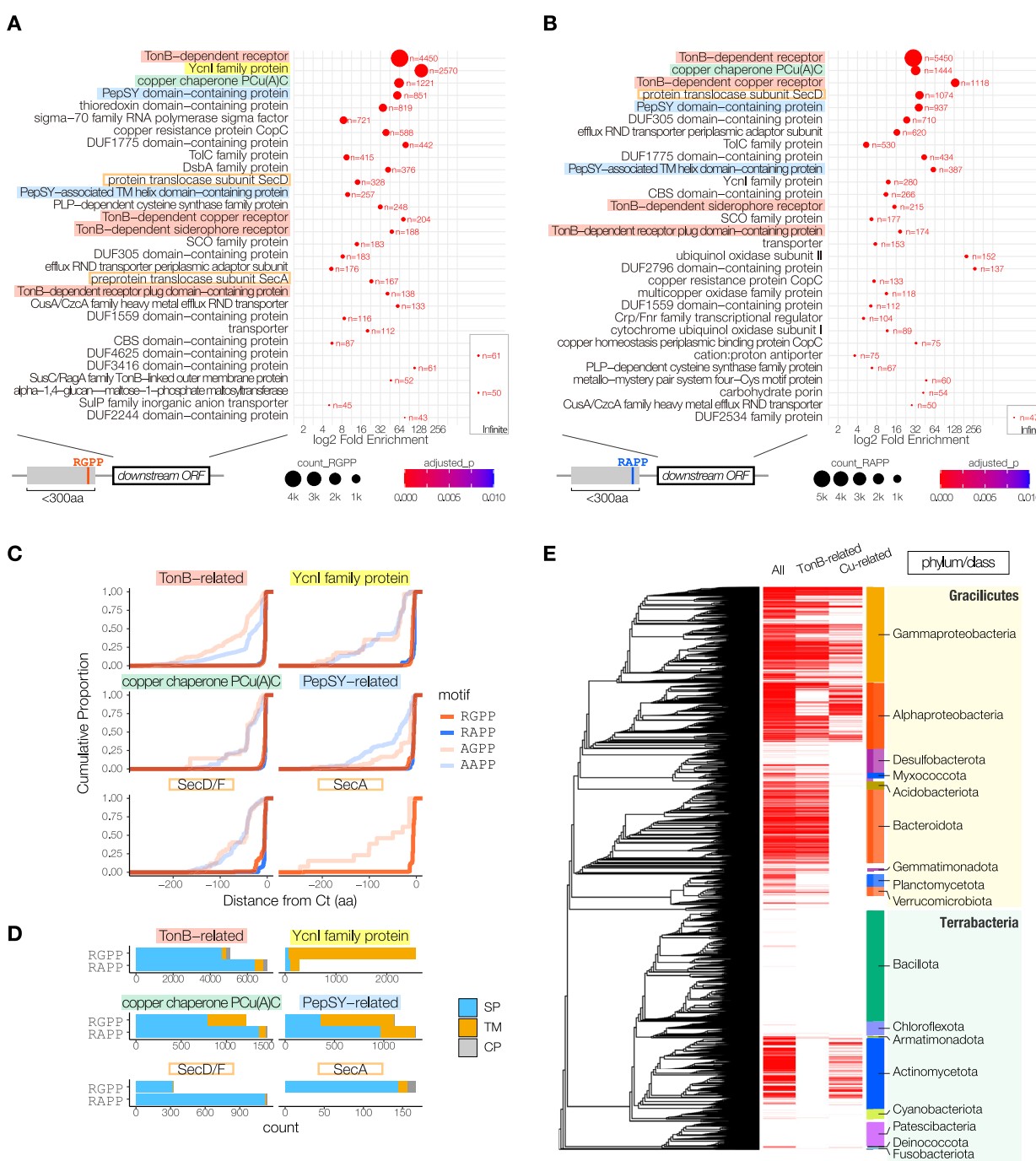

Figure 4. General features shared by small proteins with RAPP-like motif.

(A, B) Small proteins with RGPP (A) or RAPP (B) motifs are encoded upstream of genes with diverse functions. Top 30 gene groups with a higher proportion of small uORFs (<300 aa) encoding RGPP compared to AGPP, or RAPP compared to AAPP, were extracted. The horizontal axis represents fold enrichment (RGPP/AGPP or RAPP/AAPP), while the size of the circles corresponds to the number of genes in each group. Circle color indicates the p-value (Fisher's exact test) after Benjamini–Hochberg adjustment. Those for which a uORF encoding AAPP or AGPP was not found are enclosed in squares. See Appendix Fig. S2 for other groups. (C) ECDF plots show the distances between the C-terminus and RGPP (red), RAPP (blue), AGPP (pale red), or AAPP (pale blue) motifs. uORFs are categorized based on the groups of their downstream genes. See Appendix Fig. S3 for other groups. (D) Stacked bar plots show the numbers of secretory proteins (SP: light blue), transmembrane proteins (TM: light orange), and cytosolic proteins (CP: grey) containing RGPP/RAPP motifs, categorized by downstream gene groups. See Appendix Fig. S4 for additional groups. (E) Phylogenetic distribution of RGPP/RAPP-containing proteins. Those encoded upstream of genes categorized as TonB-related or Cu-related are also independently shown. Bacterial phyla or classes are represented on the right with distinct colors. Source data are available online for this figure.

receptors (Fig. 4A,B; pale red), YcnI family proteins (yellow), copper chaperone PCu(A)C (green), and PepSY domain-containing proteins (pale blue) exhibited high ratios. These genes are involved in metal homeostasis. In addition, genes encoding DsbA family proteins or thioredoxin domain-containing proteins, both of which are related to oxidative protein folding, were also identified.

We classified RAPP/RGPP-containing proteins based on the annotation of the downstream genes and individually analyzed common features shared by these proteins in each group (Fig. 4C,D). After the classification, the tendency for RAPP/RGPP motifs to occur near the C-terminus became even more pronounced (Fig. 4C; Appendix Fig. S3). This was also true for the tendency to possess protein localization signals (Fig. 4D). For example, almost all proteins encoded by uORF of TonB-dependent receptors were predicted to have a signal sequence (Fig. 4D, TonB-related), whereas most proteins encoded by uORF of the YcnI family were predicted to have transmembrane segments (Fig. 4D, YcnI-family). Similarly, many of the proteins encoded upstream of the genes listed in Fig. 4A,B were found to have a localization signal (Fig. 4D; Appendix Fig. S4).

Many arrest peptides regulate the expression of downstream genes by translation arrest near their C-terminus. If the same scenario applies to currently identified proteins with C-terminal RAPP-like motifs, it raises the possibility that a diverse range of genes—beyond those involved in the Sec or YidC pathways—might be regulated through translation arrest mechanisms. Indeed, Jacob-Dubuisson and colleagues recently demonstrated that CruR from *Bordetella pertussis*, which contains a C-terminal RAPP sequence, regulates the expression of a downstream TonB-dependent receptor in response to cellular copper concentration, in a RAPP motif-dependent manner (Roy et al, 2022).

Phylogenetic analysis revealed that RAPP/RGPP-containing peptides encoded upstream of the genes listed in Fig. 4A,B were distributed across a wide range of bacterial phyla (Fig. 4E). uORFs of the TonB-related class were universally found in Gram-negative bacteria, which has the outer membrane. uORFs of genes encoding Cu-related factors were also found in various bacterial lineages, with the exception of some groups such as Bacillota and Bacteroidota. Remarkably, excluding Actinomycetota, RGPP/RAPP-coding uORFs were absent in many of the Gram-positive bacteria including Bacillota and Cyanobacteriota. These data suggest that translation arrest and downstream gene regulation by RAPP/RGPP-containing proteins are employed by a wide range of bacteria, while also exhibiting lineage-specific preferences that might have been shaped by lineage-specific evolutionary reasons.

## Translation arrest of RAPP/RGPP-containing proteins in *Streptomyces lividans*

As a proof-of-concept, we focused on RAPP/RGPP-containing proteins from the actinomycete *Streptomyces lividans* to examine whether they indeed induce translation arrest. *S. lividans* has nine proteins containing RAPP or RGPP sequence. Seven proteins have RAPP/RGPP sequences near their C-terminal ends (Fig. 5A) and downstream genes oriented in the same direction (Appendix Fig. S5). Those include an ApdA homolog (SLIV_07330). All but one (SLIV_32905) of these seven proteins have predicted localization signals (Fig. 5A). We focused on these seven proteins and tested their ability to induce translation arrest. Due to the lack of

functional annotations, the proteins are designated by their locus tags throughout the text. We employed another hybrid PURE system, Sl PURE, in which the ribosome derived from *S. lividans* were combined with *E. coli* translation components. Reporter genes encoding GFP, the C-terminal segment of RAPP/RGPP-containing proteins, and lacZα, were translated in Ec PURE, Bs PURE, and Sl PURE systems in the presence of EF-P. The Arg mutant derivatives (AAPP and AGPP) were also translated.

Translation of wild-type derivatives of all seven proteins in Bs PURE and Sl PURE predominantly accumulated RNase-sensitive arrest products (Fig. 5B, upper and middle panels). In contrast, translation of mutant derivatives accumulated full-length species. When translated in Ec PURE, all proteins yielded full-length species as the major translation products, except for SLIV_33605, which accumulated arrest and full-length products at approximately the same levels (Fig. 5B, lower panels). These results suggest that all seven RAPP/RGPP-containing proteins induce translational arrest with high efficiency in both Sl and Bs PURE systems and that the RAPP/RGPP motif is essential for translational arrest.

The ability of these proteins to induce translation arrest was also confirmed in vivo by expressing GFP-'AP-lacZ reporters in *B. subtilis* (Fig. 5C). In all cases, wild-type reporter strains exhibited significantly lower levels of β-galactosidase activity than their Arg mutant variants, demonstrating that these seven RAPP/RGPP-containing proteins derived from *S. lividans* induce translation arrest in a RAPP/RGPP motif-dependent manner (Fig. 5C).

To gain insights into the physiological roles of RAPP/RGPP-containing arrest peptides identified in *S. lividans*, we performed in silico analysis of their N-terminal regions with respect to sequence conservation and predicted functional domains (Appendix Fig. S6). SLIV_07330, which is encoded upstream of *secDF*, exhibited neither detectable conserved sequence motifs nor AlphaFold3-predicted structured domains. In the case of SLIV_16130, whose downstream gene is related to metal homeostasis, several highly conserved residues were identified, including cysteine and histidine residues, which are often involved in metal binding. AlphaFold3 predictions suggested that two conserved cysteines (Cys48 and Cys66) are positioned in close proximity in the folded structure, potentially forming a copper-binding site (Appendix Fig. S7). Conserved histidines (His50, His58, His61) were also clustered in spatial proximity. SLIV_18480, which has another metal-related downstream gene, was predicted by deepTMHMM to contain four transmembrane segments (Fig. 5A). AlphaFold3 further suggested the formation of a copper-binding site involving three conserved histidines and one methionine (Appendix Fig. S7). In addition, Cys40 and Cys127 were predicted to be spatially adjacent.

SLIV_27375, encoded upstream of a gene for thioredoxin domain protein, was also predicted to have four putative transmembrane regions (Fig. 5A). Two conserved cysteines (Cys88 and Cys109) are located close to each other in the predicted structure, and the regions surrounding these cysteines were also highly conserved. SLIV_33595 and SLIV_33605, located upstream of genes encoding ECF σ factors, were both predicted to possess four transmembrane domains (Fig. 5A). Among these, TM1, TM3, and TM4 showed strong sequence conservation, along with the region between TM2 and TM3 and approximately 20 amino acids downstream of TM4 (Appendix Fig. S6). The functional relevance of these conserved regions remains unknown. Taken together, some of the above in silico analyses suggest functional links between the arrest peptides and their downstream genes.

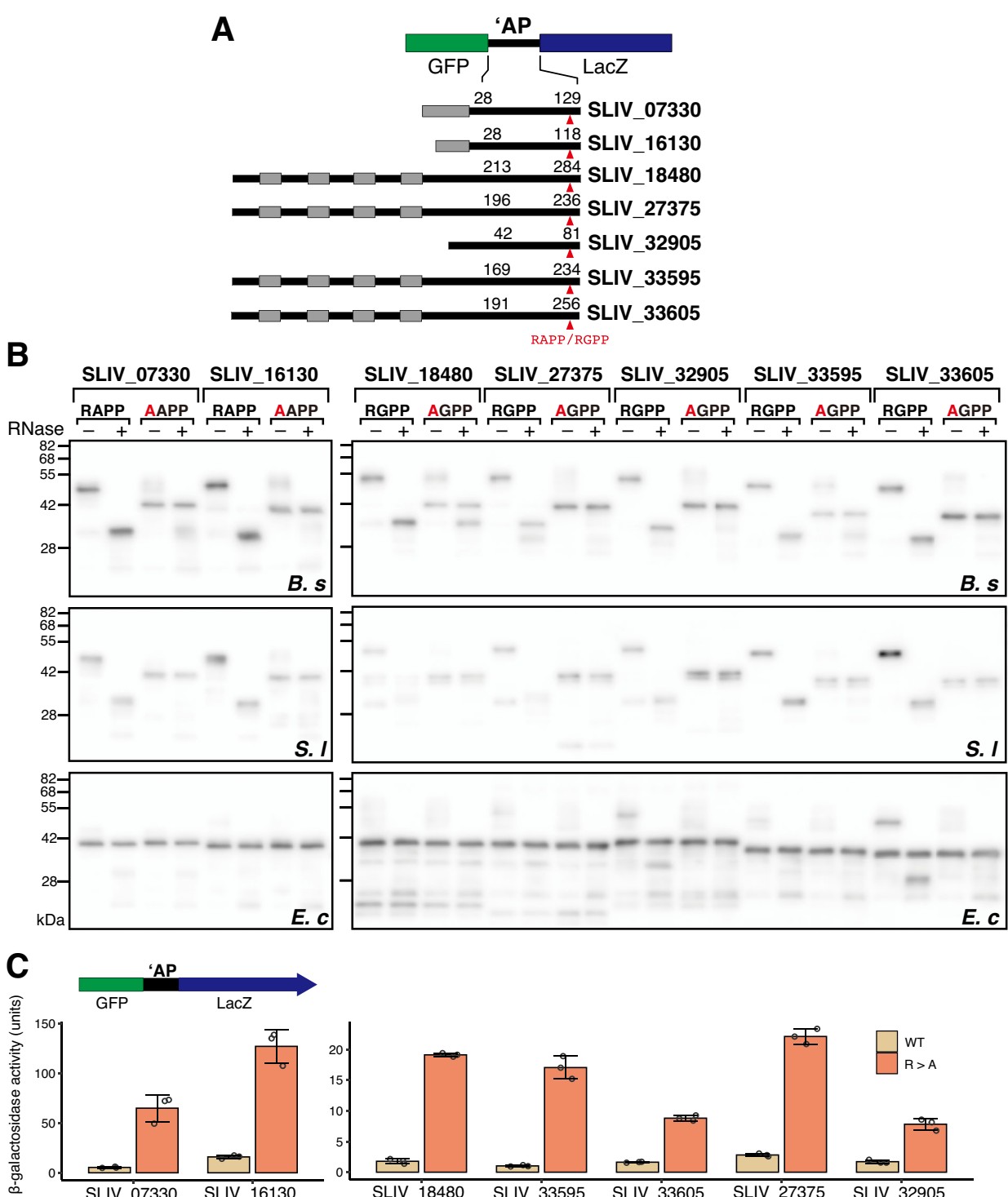

**Figure 5.  RAPP/RGPP-containing proteins from *Streptomyces lividans* induce translation arrest.**

(A) Schematic representation of the GFP-'AP-LacZα translational fusion reporter used in the in vitro translation assay. Gray boxes represent signal sequences (SLIV_07330 and SLIV_16130) or transmembrane segments (SLIV_18480, SLIV_27375, SLIV_33595 and SLIV_33605). (B) Western blot analysis of the in vitro translation products synthesized using Bs (upper), Sl (middle), and Ec (lower) PURE systems. Translation products were separated using neutral-pH gels and immunoblotted with an anti-GFP antibody. Prior to the separation, a portion of the samples was treated with RNase A (lanes indicated as +) to degrade the tRNA moiety. (C) In vivo translation arrest assay. β-galactosidase activity (mean, $n = 3$ biologically independent cell cultures) of *B subtilis* cells harboring wild-type (WT) or mutant reporter in which Arg was substituted by Ala (AAPP). Error bars and dots represent standard deviations and individual data points, respectively. Source data are available online for this figure.

As mentioned, *B. pertussis* CruR induces its downstream gene in a RAPP motif-dependent manner (Roy et al, 2022). Since the translation arrest of CruR has not been experimentally demonstrated, we investigated its arrest activity in vitro. Our in vitro translation of CruR in Ec PURE resulted in the accumulation of an RNase-sensitive arrest product, which was abolished by a mutation in the RAPP motif (Fig. EV5A,B). Furthermore, our toeprinting analysis suggested that translation arrest occurs when the third and fourth Pro codons of RAPP occupy the ribosomal P-site and A-site, respectively (Fig. EV5C). Together with the previous observation that the RAPP motif of CruR is essential for regulating its downstream gene (Roy et al, 2022), these data demonstrate that translation arrest plays a crucial role in CruR-mediated copper homeostasis.

## Discussion

Previous Cryo-EM studies have shown that the RAPP sequences in ApdA and ApdP, as well as the RAGP sequence in SecM, adopt an identical conformation, in which the arginine residue engages in extensive interactions with ribosomal residues near the PTC, and a specific interaction between the second alanine and the fourth proline was suggested to play a critical role in the ribosome stalling (Morici et al, 2024; Gersteuer et al, 2024). In contrast, their N-terminal adjacent sequences lack sequence similarity and adopt distinct structures (Morici et al, 2024; Gersteuer et al, 2024). Our comprehensive mutational analysis in this study demonstrates that RAPP-like motifs allow only limited amino acid substitutions (Fig. 1). These observations suggest that strictly specific interactions near the PTC are required for adopting the arrest-inducing conformation, whereas diverse N-terminal interactions can support the PTC-proximal interaction. Thus, the arrest sequence of RAPP-containing arrest peptides consists of two modules: a highly conserved C-terminal core region and a highly variable N-terminal adjacent region. The lack of strict sequence requirements in the N-terminal region explains why RAPP-containing arrest peptides are so widespread (Fujiwara et al, 2024).

While the second alanine of the RAPP motif of ApdA and ApdP could be substituted with small amino acid residues, the corresponding alanine in the RAPG motif of ApcA was intolerant to any substitution. Given that the interaction between the second and fourth residues of the RAPP motif in ApdA and ApdP plays a critical role in arrest (Morici et al, 2024), it is intriguing to consider that the identity of the fourth residue (Pro in RAPP vs Gly in RAPG) may have influenced the amino acid requirement at the second position.

Our previous mutational studies have shown that the RAPP-like motif alone is insufficient to induce translation arrest, and that the adjacent N-terminal region is essential for arrest activity (Sakiyama et al, 2021; Morici et al, 2024). It is plausible that the N-terminal region contributes to arrest by interacting with the ribosome in a way that reduces nascent chain fluctuation, thereby stabilizing the interaction between the C-terminal RAPP-like motif and PTC. If such a role can be achieved through general, rather than sequence-specific interactions, it may explain the tolerance for a wide variety of sequences in the N-terminal region.

Our results also suggest that the sequence of the C-terminal core region plays a significant role in determining the degree of context dependency and arrest strength. Particularly, RAPP and RGPP can induce arrest in combination with various N-terminal sequences (Fig. 2A–I). The sensitivity to mutations in uL4 and uL22 was also influenced by the sequence of the C-terminal core region (Fig. 2J). In addition, RGPP induced the most stable arrest in combination with different contexts compared to other RAPP-like sequences (Figs. 1 and 2). We assume that RAPP and RGPP sequences can maintain the arrest-inducing conformation at the PTC (Morici et al, 2024; Gersteuer et al, 2024) through relatively less extensive N-terminal interactions. These observations suggest that RAPP/RGPP sequences have the potential to readily evolve into translation-impeding sequences with a broader range of N-terminal sequence contexts.

Our bioinformatic analysis revealed that RAPP-like motifs are strongly excluded from the proteomes of many bacteria. Notably, RGPP, which was shown to induce the strongest arrest in combination with various N-terminal sequences, was identified as the most excluded four-amino-acid sequence across many bacterial phyla. These findings support the idea that RAPP-containing translation-impeding sequences, which are resistant to EF-P (Figs. 2 and EV1), can pose a risk to the translation machinery of a wide range of bacteria. Notably, the occurrence of RxPP motifs within proteome sequences is particularly reduced in Bacillota relative to other phyla (Fig. EV3B,C).

Other four-amino-acid motifs that are underrepresented in bacterial proteomes include RDPP, RSPP, RAGP (Fig. 3), and HGPP (Fig. EV3B,C). RDPP and RSPP are found in sequences of the arrest-permissive mutants of ApdA and ApdP (Fig. 1C–E). RSPP and HGPP were previously identified as part of arrest sequences by Buskirk and colleagues through random sequence screening in *E. coli* and were also reported to be underrepresented in bacterial proteomes (Tanner et al, 2009; Woolstenhulme et al, 2013). In addition, some of these motifs were found in homologs of naturally occurring arrest peptides identified in previous studies (Fujiwara et al, 2024; Sakiyama et al, 2021). Particularly, RAGP is a part of the SecM arrest sequence (Nakatogawa and Ito, 2002).

Although many RAPP/RGPP-containing proteins share common features with known Sec/YidC-related arrest peptides, they are encoded upstream of genes involved in pathways beyond protein localization. Examples include factors involved in metal homeostasis and disulfide bond formation (Fig. 4). Consistent with this observation, seven RAPP/RGPP-containing arrest peptides identified from *S. lividans* (Fig. 5) have downstream genes encoding various proteins, including SecDF, copper-resistance proteins, YcnI-like copper chaperones, thioredoxin-like proteins, and others, such as RNA polymerase sigma factors and enzymes of unknown function (Appendix Fig. S5). Our in silico analysis of these arrest peptides suggests functional links between newly identified arrest peptides and their downstream genes (Appendix Figs. S6 and S7). For example, two arrest peptides (SLIV_16130 and SLIV_18480) encoded upstream of metal-related genes contain conserved cysteine or histidine residues that were predicted by AlphaFold3 to form metal-binding sites. These findings are consistent with the hypothesis that SLIV_16130 and SLIV_18480 function as metal ion sensors, regulating the expression of their respective downstream genes. Additionally, it is noteworthy that SLIV_27375, encoded upstream of a thioredoxin domain protein, which is generally involved in disulfide bond formation, also harbors a conserved pair of cysteine residues that are predicted to be in close

proximity in the folded structure. These findings, together with the previous and current observations that CruR regulates the downstream TonB-dependent receptor in a translation arrest-dependent manner (Fig. EV5) (Roy et al, 2022), support the hypothesis that many bacteria employ RAPP-containing arrest peptides to regulate genes with diverse functions.

Many important questions regarding potential functions and mechanisms of RAPP/RGPP-containing arrest peptides remain to be addressed. For instance, how is translation arrest involved in these functions? Does arrest indeed regulate the expression of downstream genes? If so, what are the molecular mechanisms mediating this regulation? One particularly intriguing question concerns whether the force-dependent arrest release, which has been shown to occur when mechanical pulling is applied to the nascent chain (Goldman et al, 2015), also plays a significant role in functions beyond the monitoring of protein localization pathways. If this is the case, identifying the source of such pulling forces will be of considerable interest. In the cases of arrest peptides that monitor localization pathways, arrest release is triggered by a pulling force generated through direct interaction with the Sec or YidC machinery encoded by downstream genes. In contrast, for CruR, it has been proposed that arrest release is mediated by the interaction between CruR and copper ions (Roy et al, 2022). Although it is plausible that a pulling force is ultimately involved in arrest release, the mechanism by which this force is generated is not necessarily limited to direct interaction with the downstream gene product.

Our previous analyses showed that neither ApdP nor SecM induces translation arrest in rabbit reticulocyte lysates (Morici et al, 2024; Gersteuer et al, 2024), suggesting that eukaryotic translation machinery may have evolved to overcome RAPP-containing sequences either by the ribosome itself or with the help of *trans*-acting factors. In contrast, the elimination of RAPP-like motifs from a wide range of bacterial proteomes suggests that many bacterial species share a common "imperfection" in their translation machinery—the inability to translate many RAPP-containing proteins, despite having evolved various translation enhancing *trans*-acting factors, such as EF-P and ABCF family proteins (Doerfel et al, 2013; Ude et al, 2013; Peil et al, 2013; Takada et al, 2024b, 2024a; Hong et al, 2024; Ousalem et al, 2024; Chadani et al, 2024). Nevertheless, our results suggest that bacteria have actively repurposed translation-stalling sequences into various regulatory mechanisms to maintain cellular homeostasis. This underscores the remarkable flexibility of bacterial evolution, allowing them to transform the intrinsic imperfection of their translation machinery into functional advantages.

# Methods

### Reagents and tools table

| Reagent/resource | Reference or source | Identifier or catalog number |
|---|---|---|
| **Experimental models** | | |
| PY79 (*Bacillus subtilis*) | Youngman et al (1984) | N/A |
| JM109 (*Escherichia coli*) | Takara | 9052 |
| TK24 (*Streptomyces lividans*) | Dr. Ohnishi and Dr. Tezuka | N/A |

| Reagent/resource | Reference or source | Identifier or catalog number |
|---|---|---|
| **Recombinant DNA** | | |
| *apcA* | Sakiyama et al (2021) | N/A |
| *apdA* | Sakiyama et al (2021) | N/A |
| *apdP* | Sakiyama et al (2021) | N/A |
| **Antibodies** | | |
| Mouse anti-GFP | Fujifilm Wako | mFX75 |
| Rabbit anti-LacZ | Fujiwara et al (2024) | N/A |
| Goat anti-mouse IgG-HRP | Biorad | 1706516 |
| **Oligonucleotides and other sequence-based reagents** | | |
| PCR primers | This study | Appendix Table S4 |
| **Chemicals, enzymes, and other reagents** | | |
| PrimeSTAR GXL | Takara | R050B |
| T5 exonuclease | NEB | M0663 |
| Phusion DNA polymerase | NEB | M0530 |
| Taq DNA ligase | NEB | M0208 |
| DpnI | Takara | 1235B |
| Sera-Mag Carboxylate-Modified Magnetic Particles | Cytiva | 65152105050250 |
| Tryptone | Nacalai | 35640-95 |
| Yeast extract | Nacalai | 15838-45 |
| Ampicillin | Nacalai | 19796-22 |
| Chloramphenicol | Nacalai | 08027-72 |
| Y-PER | Thermo Fisher | 78990 |
| ONPG | Nacalai | 25027-71 |
| PUREfrex version 1.0 | GeneFrontier | PF001 |
| T7 RNA polymerase | Takara | 2540A |
| RNase A | Promega | A7973 |
| WIDE RANGE Gel buffer | Nakalai | 07831-94 |
| PVDF membrane | Merck | IPVH00010 |
| PefaBlock | Roche | 11429876001 |
| Leupeptin | Roche | 11034626001 |
| Pepstatin | Roche | 11359053001 |
| HiTrap Butyl FF column | GE | 17519701 |
| EF-P | GeneFrontier | PFS052 |
| dNTP mixture | Takara | 4030 |
| ReverTra Ace | Toyobo | TRT-101 |
| NTC buffer | Macherey-Nagel (Takara) | 740654.100 (U0654A) |
| NucleoSpin Gel and PCR Clean-up kit | Macherey-Nagel (Takara) | 740609 (U0609C) |
| HiDi formamide | Thermo Fisher | 4311320 |
| GeneScan 500 LIZ | Thermo Fisher | 4322682 |

| Reagent/resource | Reference or source | Identifier or catalog number |
|---|---|---|
| Thermo Sequenase Dye PrimerManual Cycle Sequencing Kit | Thermo Fisher | 79260 |
| Thermo Sequenase Cycle Sequencing Kit | Thermo Fisher | 785001KT |
| Thermo Sequenase DNA Polymerase | Cytiva | E79000Y |
| ECL Prime Western blotting detection kit | Cytiva | RPN2236 |
| Bacto Peptone | Gibco | 211677 |
| Bacto Malt extract | Gibco | 218630 |
| Sucrose | Nacalai | 30404-74 |
| Glucose | Nacalai | 16806-25 |
| **Software** | | |
| Adobe Illustrator | Adobe | |
| AlphaFold3 | https://alphafoldserver.com/welcome | |
| ape | https://cran.r-project.org/web/packages/ape/index.html | |
| biostrings | https://bioconductor.org/packages/release/bioc/html/Biostrings.html | |
| deepTMHMM | https://dtu.biolib.com/DeepTMHMM | |
| GeneMapper Software ver 6 | Applied Biosystems | |
| ggseqlogo | https://cran.r-project.org/web/packages/ggseqlogo/index.html | |
| ggtree | https://bioconductor.org/packages/release/bioc/html/ggtree.html | |
| ImageQuant TL | Thermo Fisher | |
| MAFFT | https://mafft.cbrc.jp/alignment/server/index.html | |
| MMseqs2 | https://github.com/soedinglab/MMseqs2 | |
| NCBI datasets | https://www.ncbi.nlm.nih.gov/datasets/docs/v2/command-line-tools/ | |
| patchwork | https://cloud.r-project.org/web/packages/patchwork/index.html | |
| phytools | https://cran.r-project.org/web/packages/phytools/index.html | |
| R | https://www.r-project.org/ | |
| rtracklayer | https://bioconductor.org/packages/devel/bioc/html/rtracklayer.html | |
| seqkit | https://bioinf.shenwei.me/seqkit/ | |
| tidyverse | https://www.tidyverse.org/ | |

| Reagent/resource | Reference or source | Identifier or catalog number |
|---|---|---|
| **Other** | | |
| Amersham Imager 600 | GE | |
| Multiskan SkyHigh | Thermo Fisher | |
| Microfluidizer LV1 | Microfluidics | |
| SeqStudio genetic analyzer | Thermo Fisher | |

## Bacterial strains and plasmids

The *E. coli* strain JM109 transformed with a plasmid encoding a *lacZ* reporter was used for β-galactosidase assay. *B. subtilis* strains, *E. coli* strains, plasmids, and DNA oligonucleotides used in this study are listed in Appendix Tables S1, S2, S3, and S4, respectively. Plasmids were constructed by fusing PCR fragments amplified with PrimeSTAR GXL (Takara) using Gibson Assembly (Gibson et al, 2009). Sera-Mag Carboxylate-Modified Magnetic Particles (Cytiva, 65152105050250) were used to purify DNA fragments. The *B. subtilis* strains were constructed by double homologous recombination between chromosomal DNA and the plasmids introduced into *B. subtilis* competent cells. The resulting recombinant clones were validated based on their antibiotic-resistance markers. LB medium supplemented with 5 μg/ml chloramphenicol was used for the selection of *B. subtilis* transformants.

## Culture media and growth conditions

*B. subtilis* cells were cultured in LB medium. *E. coli* cells harboring a *lacZ* reporter plasmid were cultured in LB medium supplemented with 100 μg/ml ampicillin, and those harboring both *lacZ* reporter and *rplV* plasmids were cultured in LB medium supplemented with both 100 μg/ml ampicillin and 20 μg/ml chloramphenicol. Cells were cultured at 37 °C and collected for Western blotting or β-galactosidase activity assay when they reached an optical density of 0.5–1.0 at 600 nm ($OD_{600}$).

## In vivo β-galactosidase assay

The β-galactosidase assay was performed as described previously (Fujiwara et al, 2018). A 100-μL aliquot of the culture was transferred to a well in a 96-well plate, and $OD_{600}$ was recorded. The culture was mixed with 50 μL of Y-PER reagent (Thermo Fisher) for 20 min at room temperature to lyse the cells. In the case of *E. coli* cells, the culture was diluted 10- or 100-fold by Z-buffer (60 mM $Na_2HPO_4$, 40 mM $NaH_2PO_4$, 10 mM KCl, 1 mM $MgSO_4$, and 38 mM β-mercaptoethanol) before the addition of Y-PER and further subjected to freeze–thaw treatment to ensure cell disruption. Subsequently, 30 μL of *o*-nitrophenyl-β-D-galactopyranoside (ONPG) in Z-buffer was added to the cell lysate, and the $OD_{420}$ and $OD_{550}$ were measured at 28 °C every 5 min over a period of 60 min. Arbitrary units of β-galactosidase activity were calculated using the following formula: $[(1000 \times V_{420} - 1.3 \times V_{550})/OD_{600}]$ where $V_{420}$ and $V_{550}$ are the first-order rate constants, $OD_{420}$/min and $OD_{550}$/min, respectively.

## In vitro translation and western blotting

Bacterial reconstituted transcription–translation coupling systems (Shimizu et al, 2001) were used in the in vitro translation assay. Specifically, for in vitro translation using *E. coli* ribosomes, we utilized PUREfrex version 1.0 (GeneFrontier) according to the manufacturer's protocol. For Bs and Sl PURE, purified *B. subtilis* and *S. lividans* ribosomes were used at the final concentration of 1 µM in the PURE system, without adding *E. coli* ribosomes. 2.5 U/µL of T7 RNA polymerase (Takara) was added to enhance transcription. The in vitro translation reaction was primed using the DNA templates listed in Appendix Table S5. The translation reaction was carried out for 30 min at 37 °C and was stopped by adding 2 × SDS–PAGE loading buffer. A portion of the sample was further treated with 0.2 mg/ml RNase A (Promega) at 37 °C for 10 min to degrade the tRNA moiety of the peptidyl-tRNA. The translation products were separated on a 10% polyacrylamide gel that was prepared using WIDE RANGE Gel buffer (Nacalai Tesque) according to the manufacturer's instructions, then transferred onto a PVDF membrane (Merck, IPVH00010) and subjected to immuno-detection using antibodies against GFP (mFX75; Wako) or LacZ (Fujiwara et al, 2024). Images were acquired and analyzed using an Amersham Imager 600 luminoi-mager (GE Healthcare), and the band intensity was quantified using ImageQuant TL (GE Healthcare).

## Purification of *B. subtilis* and *S. lividans* ribosomes

*B. subtilis* strain PY79 was grown at 37 °C in LB medium until mid-log phase. Cells were then harvested by centrifugation, washed twice with buffer I-H (10 mM HEPES-KOH pH 7.6, 15 mM Mg(OAc)$_2$, 1 M KCl, 5 mM EDTA, 10 mM β-mercaptoethanol), then once with buffer II-H (Buffer I-H with 50 mM KCl) and stored as a pellet at −80 °C. They were thawed in suspension buffer (10 mM HEPES-KOH pH 7.6, 50 mM KCl, 10 mM Mg(OAc)$_2$, 7 mM β-mercaptoethanol) with protease inhibitors (1 mg/ml Pefa-Block,10 µg/ml Leupeptin, 0.7 µg/ml Pepstatin) and disrupted by passing through a microfluidizer LV1 (Microfluidics) at 16,000 psi. Cell lysate was mixed with an equal volume of suspension buffer with 3 M ammonium sulfate and kept on ice for 30 min. Cell debris and protein precipitates were removed by centrifugation (4 °C, 10,000× *g*, for 30 min). The supernatant was subjected to HiTrap Butyl FF column of 10 ml volume (GE) equilibrated with buffer A (20 mM HEPES-KOH pH 7.6, 1.5 M (NH$_4$)$_2$SO$_4$, 10 mM Mg(OAc)$_2$, 7 mM β-mercaptoethanol). The column was then washed with 40 ml of buffer B (20 mM HEPES-KOH pH 7.6, 10 mM Mg(OAc)$_2$, 7 mM β-mercaptoethanol) containing 1.2 M ammonium sulfate and proteins were eluted with a 1.2 M to 0 M gradient of ammonium sulfate in buffer B. Fractions containing the ribosome were combined and loaded on the top of an equal volume of 30% sucrose cushion buffer (20 mM HEPES-KOH pH 7.6, 10 mM Mg(OAc)$_2$, 30 mM NH$_4$Cl, 30% sucrose, 7 mM β-mercap-toethanol) and ultracentrifuged (4 °C, 100,000× *g* for 16 h) to sediment the ribosomes. The pellet was suspended with ribosome buffer (20 mM HEPES-KOH pH 7.6, 30 mM KCl, 6 mM Mg(OAc)$_2$, 7 mM β-mercaptoethanol) and stored at −80 °C until use for in vitro translation assays. *S. lividans* ribosome was also purified according to the above procedure with some modifications. *S. lividans* TK24 cells were cultured in YEME medium (34% sucrose,

1% glucose, 0.5% peptone, 0.3% yeast extract, 0.3% malt extract, 0.04% MgCl$_2$·6H$_2$O, pH 7.0) at 30 °C and harvested at the log phase for ribosome purification.

## Toeprinting assay

In vitro translation was carried out using the Ec PURE system at 37 °C for 20 min in the presence or absence of 0.1 mg/mL chloramphenicol. The translation reaction mixture was then mixed with the same volume of the reverse transcription mixture containing 50 mM HEPES-KOH, pH 7.6, 100 mM potassium glutamate, 2 mM spermi-dine, 13 mM magnesium acetate, 1 mM DTT, 2 µM of oligonucleotide labeled with 6-carboxyfluorescein (6-FAM) at the 5' end (5'-AACGACGGCCAGTGAATCCGTAATCATGGT-3', Invitrogen), 50 µM each dNTP, and 10 U/µL ReverTra Ace (Toyobo), then incubated further at 37 °C for 15 min. The reaction mixture was diluted fivefold with the NTC buffer (Macherey-Nagel), and the reverse transcription products were purified using a NucleoSpin Gel and PCR Clean-up kit (Macherey-Nagel). The reverse transcription products were eluted with 30 µL of HiDi formamide (Thermo Fisher). Samples were then mixed with 10 µL of tenfold-diluted GeneScan 500 LIZ dye size standard (Thermo Fisher, 4322682), then heated at 96 °C for 3 min just before capillary electrophoresis. The dideoxy DNA samples used as size markers for sequencing were prepared using a Thermo Sequenase Dye Primer Manual Cycle Sequencing Kit (Thermo, 79260), Thermo Sequenase Cycle Sequencing Kit (Thermo, 785001KT), or Thermo Sequenase DNA Polymerase (Cytiva, E79000Y), according to the manufacturer's instructions, with some modifications. The DNA polymerase reaction was carried out using the same sets of template DNA and primer used for the toeprint assay. Each reaction mixture contained 0.44 µM of the 6-FAM-labeled primer, 60 µM each deoxynucleotide triphosphate (dATP, dCTP, dGTP, and dUTP), and 0.6 µM dideoxynucleotide triphosphate (either ddATP, ddCTP, ddGTP, or ddUTP). The sequencing products were purified using Sera-Mag Speed Beads and eluted with HiDi formamide. Next, 2 µL of a tenfold-diluted GeneScan 500 LIZ dye size standard was added. The toeprinting product was further diluted 1:80 before electrophoresis using HiDi formamide. The toeprinting and dideoxy sequencing products were then subjected to fragment analysis on a SeqStudio genetic analyzer (Thermo Fisher). Fragment data were analyzed and visualized using the GeneMapper software version 6 (Applied Biosystems), and processed further using Adobe Illustrator. The signals obtained from dideoxy sequencing were colored green (A), blue (C), black (G), and red (T), and then superposed for presentation.

## Occurrence frequency of four-amino-acid-sequence patterns

Genomic identifiers (Refseq ID and Genbank ID, Source Data for Fig. EV2) of representative bacterial genomes of the Genome Taxonomy Database (GTDB, release 214) (Parks et al, 2018) were collected and used to download files of genomic sequences, protein sequences, and GFF annotation files from the NCBI by using datasets command line tool (version 16.11.0) with dehydrate and rehydrate commands (O'Leary et al, 2024).

The occurrence count for each four-amino-acid sequence pattern in the predicted proteome was obtained by extracting four consecutive residues from the sequence data using a sliding window of four

characters and aggregating the results. When a phylum contained more than 1000 species, we analyzed 1000 randomly selected genomes from that phylum (Source Data for Fig. EV2). Subsequently, the occurrence frequency for each pattern was calculated by dividing the observed count for each pattern by the total number of patterns. In parallel, the expected frequency for each four-amino-acid sequence pattern was calculated by multiplying the amino acid occurrence frequencies in each predicted proteome, determined using the alphabetFrequency function of the R-package Biostrings version 2.72.1 (Pagès et al, 2025). The frequency bias was then calculated by dividing the observed frequency by the expected frequency. The occurrence bias across various species was aggregated by phylum or class and represented using the cumulative distribution function. In addition, the mean or median of the occurrence bias value for each phylum or class was calculated. Based on these results, the 20 patterns with the lowest mean values were selected and visualized in a heatmap.

## RGPP or RAPP-coding ORF extraction

From over 168,430,932 non-redundant proteins predicted from 52,895 representative genomes of the GTDB, we extracted protein sequences containing RGPP, RAPP, AGPP, or AAPP motifs, respectively, using seqkit version 2.6.1 (Shen et al, 2024). Then, we selected proteins shorter than 300 amino acids and extracted the annotations for both their coding genes and their downstream genes from the GFF files. Subsequently, we selected RGPP-, RAPP-, AGPP- or AAPP-encoding ORFs whose product function was annotated as "unknown", "hypothetical", or "uncharacterized", and further analysis was conducted on the products of the downstream genes with predicted functions, excluding those annotated as "unknown". For enrichment analysis of downstream ORFs, we conducted Fisher's Exact Test using the R program to compare the downstream genes of RGPP-encoding ORFs with those of AGPP-encoding ORFs, and those of RAPP-encoding ORFs with those of AAPP-encoding ORFs. Multiple testing correction was applied using the Benjamini–Hochberg method (Benjamini and Hochberg, 1995), and results with adjusted *p*-values below 0.01 were considered significant. Localization signal and transmembrane region were predicted by using DeepTMHMM version 1.0.24 (Hallgren et al, 2022). The gene list for RAPP- or RGPP-containing uORFs is provided as Source Data for Fig. 4A,B.

## Investigation of domain conservation

Protein sequences obtained as described above were grouped based on their corresponding downstream genes and clustered using MMseqs2 version 17-b804f (Steinegger and Söding, 2017). Sequences within each cluster were then aligned using MAFFT version v7.526 (Katoh and Standley, 2013). Conserved residues were visualized with the R-package ggseqlogo version 0.2 (Wagih, 2017).

## Phylogenetic analysis

The phylogenetic tree downloaded from the GTDB (release 214) was pruned to retain only the 52,895 genomes used for the above analysis by ape version 5.6.2 (Paradis and Schliep, 2019). The tree was visualized and annotated by ggtree version 3.14.0 (Yu et al, 2017).

## Data availability

This study includes no data deposited in external repositories.

The source data of this paper are collected in the following database record: biostudies:S-SCDT-10_1038-S44318-025-00651-6.

## Peer review information

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

## Acknowledgements

We thank Dr. Yasuo Ohnishi, and Dr. Takeaki Tezuka for providing the S. lividans strain and for their technical advice, Machiko Murata, and Naoko Muraki for their technical support. This work was supported by JSPS Grant-in-Aid for Scientific Research (Grant No. 26116008, 20H05926, 21K06053, and 25K02230 to SC, and 19K16044, 21K15020 to KF), JST, PRESTO (JPMJPR24ND to KF), Takeda Science Foundation (to SC), Institute for Fermentation, Osaka (grant G-2021-2-063 to SC, and grant G-2024-2-071 to KF), and Support for Pioneering Research Initiated by the Next Generation (SPRING) program by JST (JPMJSP2157 to NT).

## Author contributions

**Keigo Fujiwara**: Conceptualization; Data curation; Software; Formal analysis; Supervision; Funding acquisition; Investigation; Visualization; Writing—original draft; Writing—review and editing. **Naoko Tsuji**: Conceptualization; Investigation; Visualization; Writing—original draft; Writing—review and editing. **Karen Sakiyama**: Investigation; Writing—review and editing. **Hironori Niki**: Supervision; Writing—review and editing. **Shinobu Chiba**: Conceptualization; Supervision; Funding acquisition; Visualization; Writing—original draft; Project administration; Writing—review and editing.

Source data underlying figure panels in this paper may have individual authorship assigned. Where available, figure panel/source data authorship is listed in the following database record: biostudies:S-SCDT-10_1038-S44318-025-00651-6.

## Disclosure and competing interests statement

The authors declare no competing interests.

# Expanded View Figures

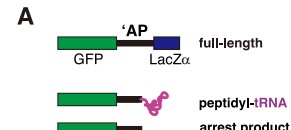

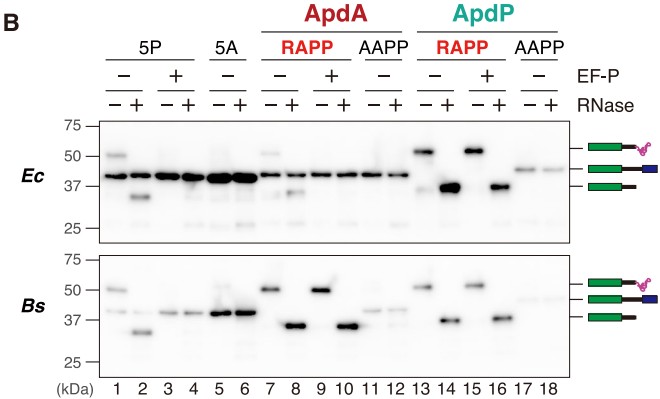

**Figure EV1.   Translation arrest of ApdA and ApdP is refractory to EF-P.**

(**A**) A schematic representation of the reporter used for the in vitro translation arrest assay. A gene segment for a protein containing five consecutive prolines (5P), alanines (5A), or the arrest sequence of ApdA or ApdP was sandwich-fused with *gfp* and *lacZα*. The tRNA is shown in magenta. (**B**) Western blot analysis of the in vitro translation products. The reporter genes harboring the 5P, 5A sequences, or the arrest motifs of wild-type (RAPP) or mutant (AAPP) sequences indicated at the top were translated in the Ec (upper) or Bs (lower) PURE systems in the presence or absence of EF-P. The products were separated in neutral-pH gels and immunoblotted using anti-GFP antibody. Before the separation, a portion of the samples were treated with RNase A, to degrade the tRNA moiety. Molecular size standards are indicated on the left. Source data are available online for this figure.

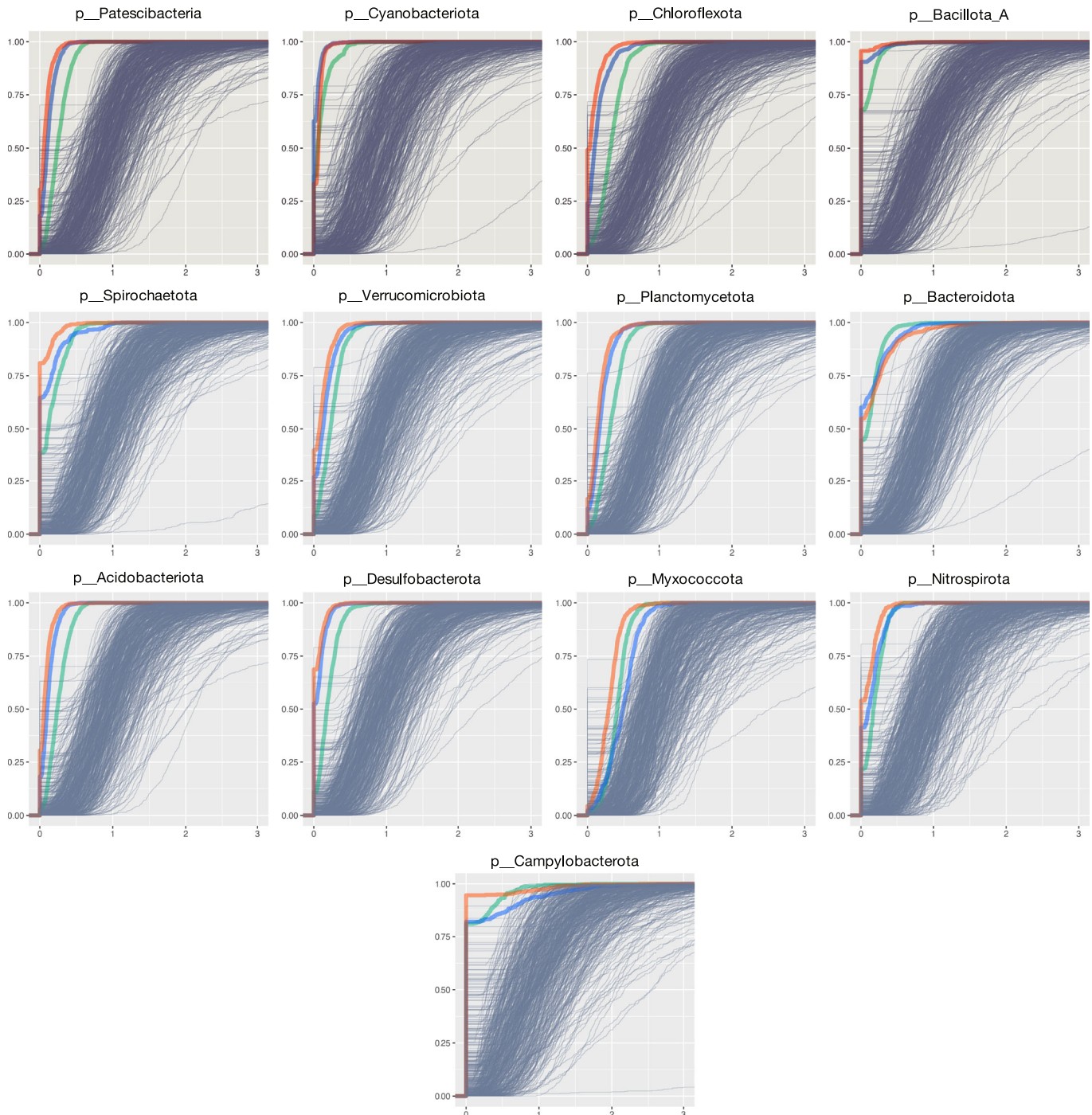

**Figure EV2. RAPP-like sequences are underrepresented across bacterial phyla.**

Empirical cumulative distribution functions (CDFs) of frequency bias for four-amino-acid sequences in phyla Patescibacteria ($n = 1000$), Cyanobacteriota ($n = 944$), Chloroflexota ($n = 1000$), Bacillota_A ($n = 1000$), Spirochaetota ($n = 606$), Verrucomicrobiota ($n = 865$), Planctomycetota ($n = 1000$), Bacteroidota ($n = 1000$), Acidobacteriota ($n = 758$), Myxococcota ($n = 425$), Nitrospirota ($n = 300$), Desulfobacterota ($n = 836$) and Campylobacterota ($n = 501$). Frequency bias values for the RGPP (red), RAPP (blue), RAGP (green) and randomly selected 500 patterns are plotted. Source data are available online for this figure.

## A

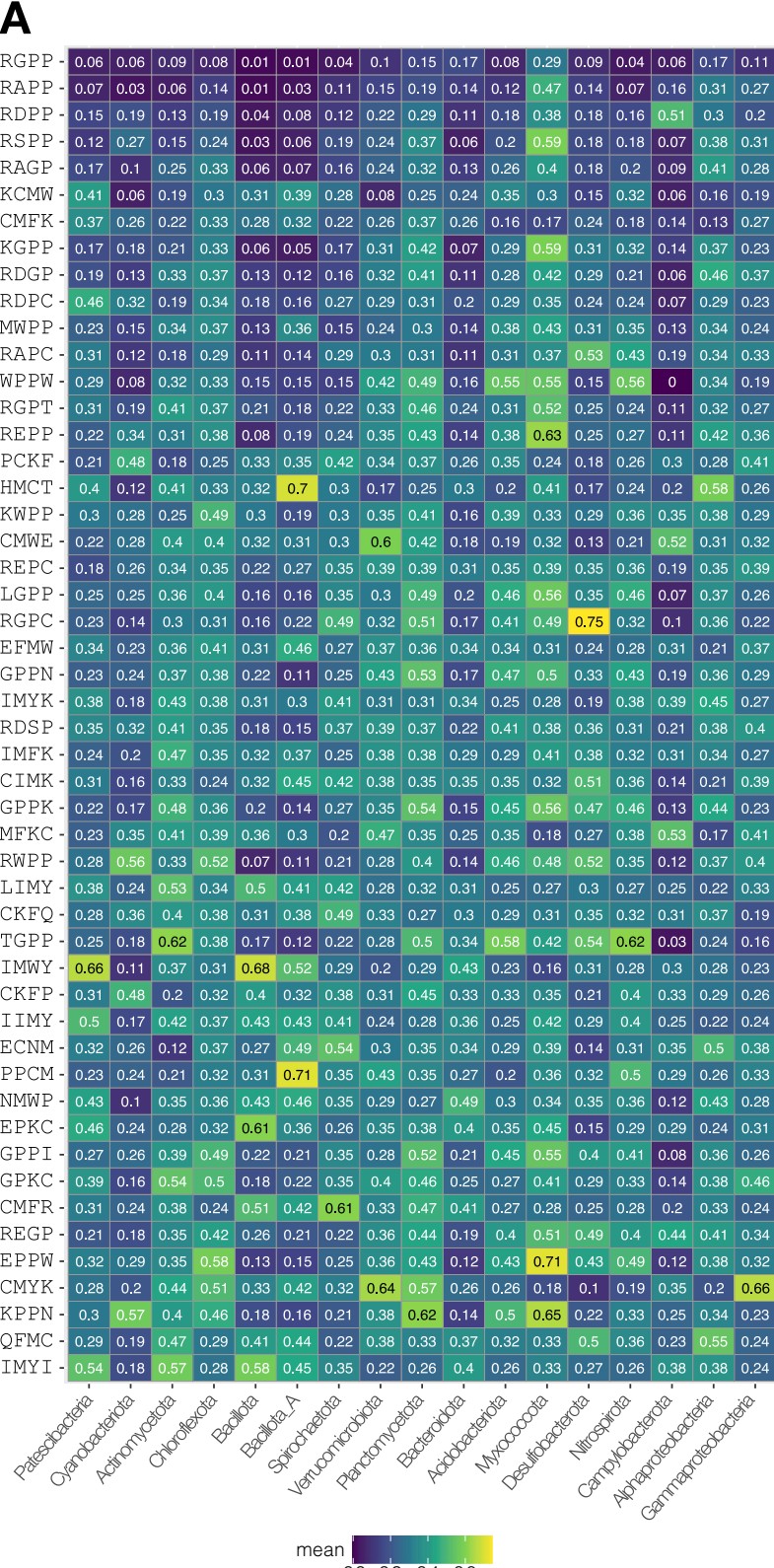

## B

Gammaproteobacteria

| pattern | mean | median | sd |
|---|---|---|---|
| RGPP | 0.1052 | 0.0712 | 0.1261 |
| CPCM | 0.1114 | 0.0000 | 1.0029 |
| HMCQ | 0.1486 | 0.0000 | 0.8045 |
| TGPP | 0.1570 | 0.1047 | 0.2622 |
| WCIM | 0.1578 | 0.0000 | 0.9353 |
| MWYD | 0.1771 | 0.0000 | 0.6419 |
| HMKC | 0.1793 | 0.0000 | 0.9448 |
| CPYM | 0.1813 | 0.0000 | 0.7585 |
| QCMC | 0.1813 | 0.0000 | 1.4293 |
| HGPP | 0.1826 | 0.0000 | 0.2616 |
| CMWY | 0.1835 | 0.0000 | 1.4602 |
| KCQM | 0.1869 | 0.0000 | 0.7010 |
| CKFQ | 0.1888 | 0.0000 | 0.5779 |
| KCMW | 0.1888 | 0.0000 | 1.4757 |
| KMCW | 0.1930 | 0.0000 | 1.2579 |
| WPPW | 0.1931 | 0.0000 | 0.6024 |
| PCMW | 0.1957 | 0.0000 | 1.2009 |
| KCMY | 0.1968 | 0.0000 | 0.9455 |
| RDPP | 0.2002 | 0.1793 | 0.1937 |
| GPPH | 0.2015 | 0.1269 | 0.3362 |

## C

Bacillota

| pattern | mean | median | sd |
|---|---|---|---|
| RGPP | 0.0107 | 0.0000 | 0.1538 |
| RAPP | 0.0144 | 0.0000 | 0.0984 |
| RSPP | 0.0302 | 0.0000 | 0.0937 |
| RDPP | 0.0373 | 0.0000 | 0.1595 |
| RPPP | 0.0429 | 0.0000 | 0.3477 |
| KGPP | 0.0619 | 0.0000 | 0.1411 |
| RAGP | 0.0632 | 0.0000 | 0.1923 |
| RWPP | 0.0743 | 0.0000 | 0.3424 |
| REPP | 0.0826 | 0.0000 | 0.1910 |
| KPPP | 0.0848 | 0.0000 | 0.1897 |
| KPPW | 0.0880 | 0.0000 | 0.3067 |
| GPPW | 0.0910 | 0.0000 | 0.3733 |
| HPPP | 0.0969 | 0.0000 | 0.3308 |
| RAPC | 0.1133 | 0.0000 | 0.4868 |
| KDPP | 0.1239 | 0.0000 | 0.2014 |
| HGPP | 0.1250 | 0.0000 | 0.4376 |
| MWPP | 0.1255 | 0.0000 | 0.6787 |
| EPPW | 0.1328 | 0.0000 | 0.5259 |
| RDGP | 0.1335 | 0.0000 | 0.2537 |
| KPPD | 0.1356 | 0.0000 | 0.2061 |

◀ **Figure EV3.  Four-amino-acid sequences underrepresented in bacteria.**

(A) Heatmap showing the average frequency bias of four-amino-acid sequences with low frequency bias. The average frequency bias for each four-amino-acid sequence was calculated for each 15 phylum and two major class of the phylum Pseudomonadota. The top 50 patterns with the lowest mean of the frequency bias across all phyla are shown. (B, C) List of the top 20 underrepresented four-amino-acid sequences for class Gammaproteobacteria (B) and phylum Bacillota (C). Source data are available online for this figure.

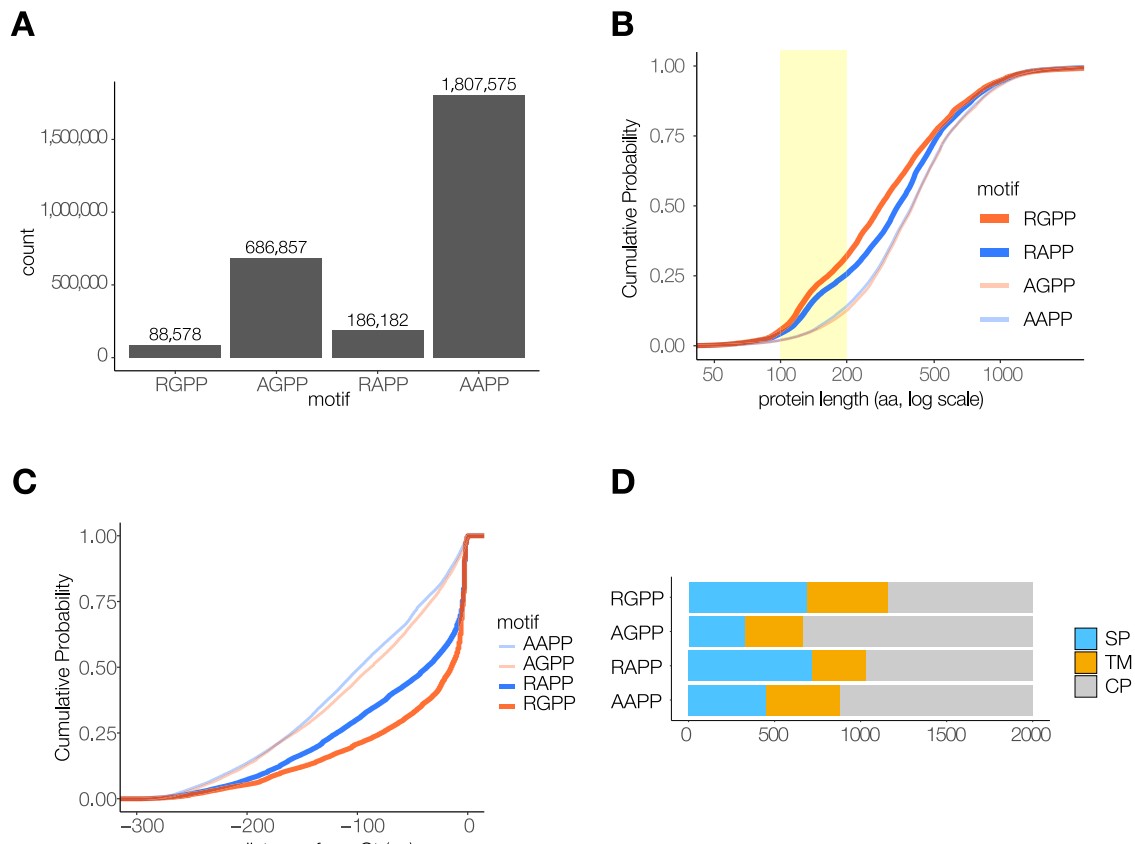

**Figure EV4.   RAPP-like motifs are enriched in the C-terminal regions of small proteins.**

(**A**) Bar chart shows the count of proteins containing RGPP, AGPP, RAPP or AAPP motif within their sequences, found among representative proteomes of bacteria. (**B**) RAPP and RGPP motifs are enriched in relatively small proteins. ECDF plots show the length distribution of proteins containing RGPP (red), RAPP (blue), AGPP (pale red), and AAPP (pale blue). (**C**) ECDF plot shows the length distribution of distance from the C-terminus and RGPP (red, $n = 46,906$), RAPP (blue, $n = 78,810$), AGPP (pale red, $n = 216,471$), and AAPP (pale blue, $n = 600,525$). Analysis was performed on proteins with 300 or fewer amino acid residues. (**D**) RAPP and RGPP motifs are enriched in secretory proteins. Stacked bar plots display the proportions of predicted secretory proteins (SP: light blue), membrane proteins (TM: orange), and cytoplasmic proteins (CP: grey) containing RGPP, RAPP, AGPP, and AAPP motifs. Analysis was performed on 2000 randomly selected proteins with 300 or fewer amino acid residues using deepTMHMM. Source data are available online for this figure.

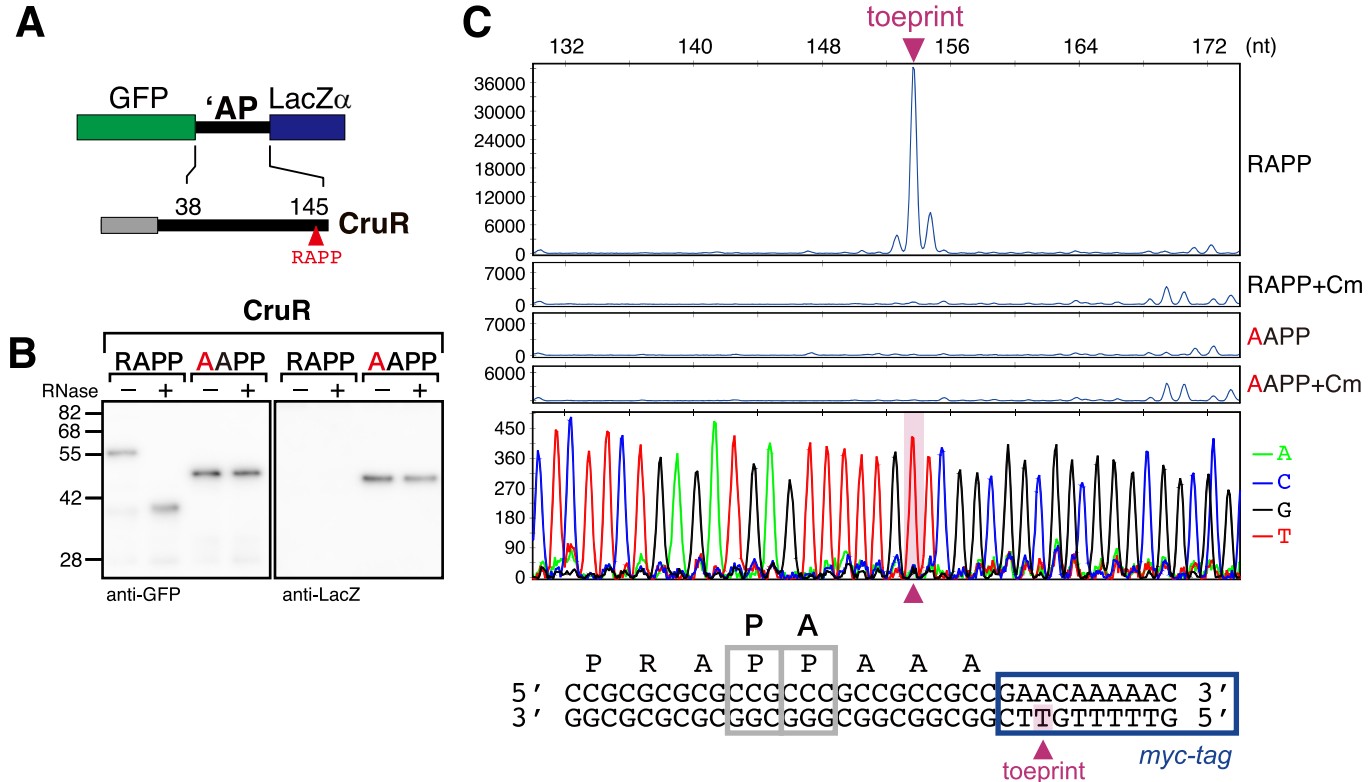

**Figure EV5.** ***Bordetella pertussis* CruR induces translation arrest in Ec PURE system.**

(A) A schematic representation of the reporter used for the in vitro translation arrest assay. A gene segment encoding the C-terminal soluble domain of CruR was fused in-frame between *gfp* and *lacZα*. (B) Western blot analysis of in vitro translation products. Since *B. pertussis* is a bacterium that belongs to the phylum Proteobacteria, we employed Ec PURE. The wild-type (RAPP) and R139A (AAPP) reporter genes, indicated at the top, were translated in the Ec PURE system in the presence of EF-P. The products were separated on neutral-pH gels and immunoblotted using anti-GFP and anti-LacZ antibodies. Prior to separation, a portion of the samples was treated with RNase A to degrade the tRNA moiety. Molecular size standards are indicated on the left. (C) Toeprint analysis of CruR. In vitro transcription–translation reaction mixtures containing templates encoding wild-type (RAPP) or R139A (AAPP) mutant reporters in the presence or absence of a translation inhibitor chloramphenicol (Cm) were subjected to fragment analysis on a SeqStudio Genetic Analyzer. Source data are available online for this figure.

