## [Peer Review File · The EMBO Journal]

Evolutionary Adaptation of Bacterial proteomes to Translation-Impeding Sequences

Keigo Fujiwara, Naoko Tsuji, Karen Sakiyama, Hironori Niki, and Shinobu Chiba

Corresponding authors: Shinobu Chiba (schiba@cc.kyoto-su.ac.jp) , Keigo Fujiwara (kig.fujiwara@nig.ac.jp)

Review Timeline:

Submission Date:	12th Apr 25
Editorial Decision:	16th May 25
Revision Received:	13th Aug 25
Editorial Decision:	16th Sep 25
Revision Received:	25th Sep 25
Accepted:	28th Oct 25

Editor: Yehu Moran

Transaction Report:

Dear Dr. Chiba,

Thank you for submitting your manuscript for consideration by the EMBO Journal. It has now been seen by three referees whose comments are shown below.

Given the referees' positive recommendations, I would like to invite you to submit a revised version of the manuscript, addressing the comments of all three reviewers. I should add that it is EMBO Journal policy to allow only a single round of revision, and acceptance of your manuscript will therefore depend on the completeness of your responses in this revised version.

I strongly suggest that together with your co-authors you will prepare a revision plan and share it with me via email in the next few weeks. This will allow us to provide you with feedback regarding your plans and from our experience might help in making the revision process smoother as it will allow both sides to coordinate their expectations.

Thank you for the opportunity to consider your work for publication. I look forward to your revision.

Yours sincerely,

Yehu Moran
Academic Editor
The EMBO Journal

We realize that it is difficult to revise to a specific deadline. In the interest of protecting the conceptual advance provided by the work, we recommend a revision within 3 months (14th Aug 2025). Please discuss the revision progress ahead of this time with the editor if you require more time to complete the revisions.

Referee #1:

This manuscript builds on a previous study where the authors found nearly 20 arrest peptides encoded upstream of genes involved in protein localisation, which was an extension of the initial discovery of SecM in *E. coli* and MifM in *B. subtilis*. This included a number of C-terminal RAP sequences causing stalled ribosomes. These sequences appear to require some other interactions through the N-terminal region but the nature of these interactions or this interplay is not known.

In this manuscript, a further detailed study is carried out employing an extensive mutagenesis of the C-terminal RAP sequences to assess the specific signature of such sequences using a reporter gene assay. Also, the N-terminal regions were swapped and combined with various RAP sequences, showing greater compositional flexibility. The remainder of the manuscript concerns a bioinformatic analysis showing that RAPP sequences are mostly avoided in protein sequences and limited to a relatively small number of proteins that not in all cases can be associated with monitoring of secretion stress of membrane protein insertion stress. The conclusion of the manuscript is that RAPP-containing arrest peptides are widespread and involved in various biological processes, but at the same time the study does not identify what kind of biological processes this concerns.

Overall, the study is well conducted, and the mutagenesis approach is quite extensive confirming the conserved nature of RAP sequences. I do not consider this a major new finding as quite some mutagenesis of such sequences has been done already also to form more stable interactions with the ribosome for specific applications such as the generation of stalled ribosomes. The combination of RAPP and RGPP sequence with different N-terminal sequences concerns some new findings, showing some interplay between the C-terminal RAP sequence and the adjoining N-terminal protein sequence. Some of this specificity is strikingly different between the four tested proteins (at least in one case with SecM using RAPG), but the key thing, i.e., the molecular basis of this specificity is not resolved in this study, nor is a plausible hypothesis provided.

The bioinformatic analysis showing that RAPP-like sequences are rare, but associated with diverse protein functions that go beyond protein secretion stress is interesting and may stimulate further work to resolve the function of such sequences in specific processes. That RAP sequences are in generally avoided in proteins is not surprising as they cause translational pausing or arresting. Hence, although the study is conducted in a sound manner, the overall insight obtained is incremental, certainly in relation to two previous publications of the group (Ref. 18/19) even though those were associated with protein localisation. What essentially is missing are new insights in their true function or why some of the newly discovered proteins require such sequences even though they are not part of protein localisation. What are these regulatory roles? I could image that this has something to do with protein complex assembly, but this really needs further work and also not immediately evident for all identified proteins. What is the molecular basis of the interplay between C-terminal RAP sequences and the protein identity? Is there any overarching testable hypothesis or explanation?

Referee #2:

The sequence of the nascent polypeptide chain that is being synthesized can modulate the function of the ribosome that is translating it. Specific sequences have been identified that can slow or even arrest translation. Such arrest peptides are usually found within short upstream open reading frames and the translation arrest is used to regulate expression of a downstream gene. Classic examples include SecM, MifM and VemP leader peptides that regulate expression of downstream genes involved in protein targeting and localization. The study here follows on previous work from the Chiba group where they identified RxPP arrest peptides (e.g. ApcA, ApdA and ApdP) located upstream of protein targeting and localization genes across a diverse range of bacterial phyla (Fujiwara Nat Comm 2024; Sakiyama NAR 2021). In collaboration, the authors also structurally resolved ApdA and ApdP-stalled ribosomes (RAPP), as well as SecM (RAGP), providing mechanistic insight into the role of the RxPP motif during translation arrest (Morici et al Nat Comm 2024; Gersteuer et al Nat Comm 2024). Here the authors extend their analysis of the ApcA, ApdA and ApdP arrest peptides to ascertain the importance of the different residues for arrest in vivo and in vitro.

They then demonstrate that such arrest motifs, including RAPP and RAGP, are highly underrepresented in bacterial genomes, whereas non-arresting variants, such as AAPP, are less heavily selected against. The first exciting discovery of the manuscript is that such RAPP and RAGP arrest peptides are nevertheless present in many bacterial genomes, located within the C-terminus of potential upstream leader peptides. The second and most important finding is that these arrest motifs are not just found upstream of genes involved in protein targeting or localization (such as SecD/F or SecA), but also in front of many other unrelated genes. This includes TonB-related, YcnI family proteins, copper chaperone and PepSY. The authors select a few examples from *S.lividens* and demonstrate using in vitro and in vivo reporter systems that these arrest peptides can stall translation.

Overall, this is a very nicely written and illustrated paper, with technically high-quality data that is appropriately interpreted and presented.

Addressing some additional points could improve the paper:

1. The first two figures dealing with the systematic characterization of the different arrest peptide motifs suggests that the ApaC arrest operates by a different mechanism than the ApdA and ApdP, however, the authors do not comment on this specifically. Moreover, it is surprising that the authors do not put their mutagenesis analysis in the context of the available structures of these staller.
2. The manuscript would benefit from additional data to either demonstrate that the leader peptides regulate expression of the downstream genes in vivo. At the moment, all the authors experiments are in vitro or using reporters, with the exception of the Cu-dependent staller that has been shown by other groups.
3. Following on from point 2, assuming there is regulation of expression in vivo, it would be pertinent to provide a model for how this regulation is working. Is the mechanism analogous to the previously characterized SecM-related mechanisms whereby a pulling force relieves the translation arrest? Many of the downstream genes are indeed membrane proteins but does this mean that they interact with the arrest peptides? Are there signal sequences or other motifs that would be recognized by the different target proteins? It is not made clear about the different localization (cytoplasmic, membrane, etc) of the different regulatory proteins and whether they are expected to be directly involved in sensing the arrested ribosomes. Is it only the RxPP motif that is conserved in these leader peptides or are there also differential conservation of the N-terminal motifs depending on the downstream genes?

Referee #3:

The paper of Fujiwara et al. expands on the previous impressive body of work from the Chiba lab on the translation arrest peptides in bacteria. Here, the authors investigated the contribution of individual amino acids of the RAPP and RAGP arrest motifs to the translation arrest and then carried out an extensive bioinformatics analysis of the occurrence of the putative arrest motifs in the bacterial genomes. This analysis suggests that the putative arrest motifs are generally counter-selected in the bacterial proteomes. However, they are retained in short proteins whose translation may regulate expression of the downstream genes with various functions.

Critique

This is an interesting work, based on the extremely 'clean' experimental data and insightful bioinformatics analysis. I have only a couple of semi-major critics. The rest are more of a cosmetic nature.

Specifically, the role of the sequences N- adjacent to the stalling RAPP and RGPP motifs are not sufficiently addressed. Thus, it remains unclear, whether RGPP and possibly some other stalling tetrapeptide sequences would stall translation irrespective of the rest of the sequence context. In their previous study (ref. 20) authors partly addressed this question for the RAPP sequence. They also mention briefly in this manuscript the possible effect of the N-terminal context. However, the reader would benefit from a more explicit discussion of whether the tetrapeptide motif is sufficient (or not) for the translation arrest.

I was also surprised why, with using a GFP-lacZ reporter, the authors did not normalize beta-galactosidase activity by GFP fluorescence. Conceivably, the mutations they introduce might affect mRNA stability; GFP-based normalization could possibly account for this.

Other comments:

l. 61: either 'about' or 'nearly' not 'about nearly'

l.99; add 'ApdA' in the parentheses as a RAPP-containing peptide.

l. 175 replace 'know' with 'known'.

l. 197. Delete 'whether'

l. 300. Decipher 'GTDB' on the first use of the abbreviation.

Fig. 1a and 5a. What do gray boxes within the GFP-lacZ intergenic region represent?

Fig. 5b. The RNase + and - signs seem to be flipped.

Responses to reviewers

General responses to the reviewers

We appreciate the reviewers' overall positive evaluation, as well as helpful and constructive comments and suggestions. We have revised the manuscript and added additional data according to a plan pre-discussed with the Academic Editor, Dr. Yehu Moran.

To gain further insight into the physiological function of the newly identified arrest peptides from *S. lividans*, we attempted additional *in vivo* reporter assays using *B. subtilis*. In addition, we also performed *in silico* analyses to explore functional links between arrest peptides and their downstream genes.

In the reporter assays, we aimed to examine whether translation arrest caused by *S. lividans* arrest peptides affects the expression of downstream genes. Unfortunately, no downstream gene expression was detected in our reporter assay, likely due to interspecies differences in gene regulatory mechanisms between *B. subtilis* and *S. lividans*.

On the other hand, the *in silico* analyses yielded several intriguing results suggesting functional relationships between the arrest peptides and their downstream genes. We believe these data provide a clue to the functions of newly identified arrest peptides and their role in the downstream gene expression, and thus offer an answer to the comments from Reviewers 1 and 2. We have included these results in the revised manuscript (in Results, Discussion, and Appendix Fig. S6, S7).

We also added a description of the toeprinting analysis in the Methods section, which had been missing from our previous version. We also corrected several minor errors and adjusted the manuscript formatting to comply with The EMBO Journal style.

Following are our point-to-point responses to the referees' comments (*in Italic*)

Referee #1:

This manuscript builds on a previous study where the authors found nearly 20 arrest peptides encoded upstream of genes involved in protein localisation, which was an extension of the initial discovery of SecM in E. coli and MifM in B. subtilis. This included a number of C-terminal RAP sequences causing stalled ribosomes. These sequences appear to require some other interactions through the N-terminal region but the nature of these interactions or this interplay is not known.

In this manuscript, a further detailed study is carried out employing an extensive mutagenesis of the C-terminal RAP sequences to assess the specific signature of such sequences using a reporter gene assay. Also, the N-terminal regions were swapped and combined with various RAP sequences, showing greater compositional flexibility. The remainder of the manuscript concerns a bioinformatic analysis showing that RAPP sequences are mostly avoided in protein sequences and limited to a relatively small number of proteins that not in all cases can be associated with monitoring of secretion stress of membrane protein insertion stress. The conclusion of the manuscript is that RAPP-containing arrest peptides are widespread and involved in various biological processes, but at the same time the study does not identify what kind of biological processes this concerns.

Overall, the study is well conducted, and the mutagenesis approach is quite extensive confirming the conserved nature of RAP sequences. I do not consider this a major new finding as quite some mutagenesis of such sequences has been done already also to form more stable interactions with the ribosome for specific applications such as the generation of stalled ribosomes.

Response: The functional significance of the RAP sequence was previously demonstrated through alanine-scanning mutagenesis (Sakiyama et al. 2021); however, the present comprehensive mutagenesis provides a substantially deeper understanding. By covering all 19 non-native amino acids at each position, this study revealed that the RGPP motif induces more stable translational arrest than the RAPP motif in both ApdA and ApdP. This finding led to the novel insight that RGPP can act as a more potent arrest sequence than RAPP. This observation provides a compelling interpretation of the bioinformatic analysis presented later in the study, which shows that RGPP motifs are

even more strongly counter-selected in bacterial proteomes compared to RAPP. We believe that integrating these observations in a unified manner underscores the biological significance of the mutagenesis conducted in this study.

The combination of RAPP and RGPP sequence with different N-terminal sequences concerns some new findings, showing some interplay between the C-terminal RAP sequence and the adjoining N-terminal protein sequence. Some of this specificity is strikingly different between the four tested proteins (at least in one case with SecM using RAPG), but the key thing, i.e., the molecular basis of this specificity is not resolved in this study, nor is a plausible hypothesis provided.

Response: As acknowledged by the reviewer, the observation that RAPP and RGPP motifs can function in combination with a variety of upstream N-terminal regions provides intriguing insights, particularly when considered alongside our bioinformatic analysis showing that such motifs are strongly counter-selected in proteomes. However, as also noted by the reviewer, its molecular basis remains unresolved in the present study. Previous structural studies (Morici et al., 2024; Gerstlauer et al., 2024) have demonstrated that RAPP and RAGP motifs adopt similar conformations near the PTC, yet the N-terminal regions of ApdA and ApdP, likely due to their inherent flexibility, could not be visualized in the structural analyses. Therefore, we are currently unable to propose a plausible hypothesis regarding the interplay between the N-terminal region and the C-terminal RAPP core motif. However, a basic idea regarding the plausible role of the N-terminal adjacent region is that the N-terminal region somehow reduces the fluctuation of the nascent chain to stabilize the arrest-essential interactions between RAP motif and the PTC. Therefore, as also mentioned in our response to Reviewer 3 (see below), we added the following statement in Discussion (line 457-461).

“It is plausible that the N-terminal region contributes to arrest by interacting with the ribosome in a way that reduces nascent chain fluctuation, thereby stabilizing the interaction between the C-terminal RAPP-like motif and PTC. If such a role can be achieved through general, rather than sequence-specific interactions, it may explain the tolerance for a wide variety of sequences in the N-terminal region.”.

The bioinformatic analysis showing that RAPP-like sequences are rare, but associated

with diverse protein functions that go beyond protein secretion stress is interesting and may stimulate further work to resolve the function of such sequences in specific processes.

Response: We fully agree with the reviewer's comment that this finding "may stimulate further work."

That RAP sequences are in generally avoided in proteins is not surprising as they cause translational pausing or arresting. Hence, although the study is conducted in a sound manner, the overall insight obtained is incremental, certainly in relation to two previous publications of the group (Ref. 18/19) even though those were associated with protein localisation. What essentially is missing are new insights in their true function or why some of the newly discovered proteins require such sequences even though they are not part of protein localisation. What are these regulatory roles? I could image that this has something to do with protein complex assembly, but this really needs further work and also not immediately evident for all identified proteins. What is the molecular basis of the interplay between C-terminal RAP sequences and the protein identity? Is there any overarching testable hypothesis or explanation?

Response: We thank the reviewer for highlighting these important questions. As pointed out by the reviewer, further studies on individual arrest peptides are necessary to clarify the functions of the newly identified arrest peptides. We believe that functional analyses of individual arrest peptides fall beyond the scope of the present study and should be addressed in a separate paper in the future.

Nevertheless, as a first step to address the above and related questions raised by Reviewer 2 (see below), we performed additional experiments to determine whether the newly identified arrest peptides in *S. lividans* can induce downstream gene expression in an arrest-dependent manner using *B. subtilis* reporter assay. We also carried out in silico analyses to explore the functions of these arrest peptides. Unfortunately, in all cases, the reporter assay failed to detect downstream gene expression, presumably due to differences in the gene regulation mechanisms between *S. lividans* and *B. subtilis* (see a file for Results of additional experiments). On the other hand, our in silico analyses revealed conserved Cys or His residues in SLIV_16130 and SLIV_18480, predicted by

AlphaFold3 to form copper-binding sites, and a conserved Cys pair in SLIV_27375 predicted to be in close proximity in the folded structure. Given that their downstream genes are related to copper homeostasis or encode a thioredoxin-like protein, these findings support the idea that arrest peptides act as sensors of metal ions or oxidative folding capacity to feedback-regulate downstream genes (lines 400-423 in Results, lines 494-502 in Discussion, and Appendix Fig. S6, 7).

Regarding the interplay between the N-terminal region and the C-terminal RAP motif, we also added a brief discussion noting the possible involvement of force-dependent arrest release (lines 506-513).

“Many important questions regarding potential functions and mechanisms of RAPP/RGPP-containing arrest peptides remain to be addressed. For instance: How is translation arrest involved in these functions? Does arrest indeed regulate the expression of downstream genes? If so, what are the molecular mechanisms mediating this regulation? One particularly intriguing question concerns whether the force-dependent arrest release, which has been shown to occur when mechanical pulling is applied to the nascent chain (Goldman *et al*, 2015) also plays a significant role in functions beyond the monitoring of protein localization pathways. If this is the case, identifying the source of such pulling forces will be of considerable interest.”

Referee #2:

The sequence of the nascent polypeptide chain that is being synthesized can modulate the function of the ribosome that is translating it. Specific sequences have been identified that can slow or even arrest translation. Such arrest peptides are usually found within short upstream open reading frames and the translation arrest is used to regulate expression of a downstream gene. Classic examples include SecM, MifM and VemP leader peptides that regulate expression of downstream genes involved in protein targeting and localization. The study here follows on previous work from the Chiba group where they identified RxPP arrest peptides (e.g. ApcA, ApdA and ApdP) located upstream of protein targeting and localization genes across a diverse range of bacterial phyla (Fujiwara Nat Comm 2024; Sakiyama NAR 2021). In collaboration, the authors also structurally resolved ApdA and ApdP-stalled ribosomes (RAPP), as well as SecM

(RAGP), providing mechanistic insight into the role of the RxPP motif during translation arrest (Morici et al Nat Comm 2024; Gersteuer et al Nat Comm 2024). Here the authors extend their analysis of the ApcA, ApdA and ApdP arrest peptides to ascertain the importance of the different residues for arrest in vivo and in vitro. They then demonstrate that such arrest motifs, including RAPP and RAGP, are highly underrepresented in bacterial genomes, whereas non-arresting variants, such as AAPP, are less heavily selected against. The first exciting discovery of the manuscript is that such RAPP and RAGP arrest peptides are nevertheless present in many bacterial genomes, located within the C-terminus of potential upstream leader peptides. The second and most important finding is that these arrest motifs are not just found upstream of genes involved in protein targeting or localization (such as SecD/F or SecA), but also in front of many other unrelated genes. This includes TonB-related, YcnI family proteins, copper chaperone and PepSY. The authors select a few examples from S.lividens and demonstrate using in vitro and in vivo reporter systems that these arrest peptides can stall translation.

Overall, this is a very nicely written and illustrated paper, with technically high-quality data that is appropriately interpreted and presented.

Response: We appreciate reviewer's positive evaluation.

Addressing some additional points could improve the paper:

1. The first two figures dealing with the systematic characterization of the different arrest peptide motifs suggests that the ApcA arrest operates by a different mechanism than the ApdA and ApdP, however, the authors do not comment on this specifically. Moreover, it is surprising that the authors do not put their mutagenesis analysis in the context of the available structures of these stagers.

Response: In accordance with the reviewer's comment, we have added the following statement regarding difference between ApcA and ApdA/ApdP in Discussion (lines 449-454).

“While the second alanine of the RAPP motif of ApdA and ApdP could be substituted with small amino acid residues, the corresponding alanine in the RAGP motif of ApcA

was intolerant to any substitution. Given that the interaction between the second and fourth residues of the RAPP motif in ApdA and ApdP plays a critical role in arrest (Morici *et al*, 2024), it is intriguing to consider that the identity of the fourth residue (Pro in RAPP vs Gly in RAPG) may have influenced the amino acid requirement at the second position.”

We also added the following discussion that integrates our mutational analysis with the previous structural studies in Discussion (line 435-439)

“Previous Cryo-EM studies have shown that the RAPP sequences in ApdA and ApdP, as well as the RAGP sequence in SecM, adopt an identical conformation, in which the arginine residue engages in extensive interactions with ribosomal residues near the PTC, and a specific interaction between the second alanine and the fourth proline was suggested to play a critical role in the ribosome stalling (Morici *et al*, 2024; Gersteuer *et al*, 2024).”

2. The manuscript would benefit from additional data to either demonstrate that the leader peptides regulate expression of the downstream genes in vivo. At the moment, all the authors experiments are in vitro or using reporters, with the exception of the Cu-dependent staller that has been shown by other groups.

Response: In accordance with the reviewer’s suggestion, and as noted in our response to Reviewer 1’s comment, we attempted to experimentally validate the arrest-dependent regulation of downstream gene expression using a *B. subtilis* reporter assay, focusing on the newly identified arrest peptides from *S. lividans*. However, as mentioned above, no downstream gene expression was observed (see also a file for Results of additional experiments). Since the genetics of *S. lividans* is more challenging than that of model organisms such as *B. subtilis*, we consider establishing such an experimental system and conducting analyses with it to be beyond the scope of the present study. Nevertheless, our in silico analyses, which were performed in response to Comment 3 from Reviewer 2 and a related question from Reviewer 1 provide further evidence for functional links between the arrest peptides and their downstream genes, partly addressing the above comment (see below).

3. Following on from point 2, assuming there is regulation of expression *in vivo*, it would be pertinent to provide a model for how this regulation is working. Is the mechanism analogous to the previously characterized SecM-related mechanisms whereby a pulling force relieves the translation arrest? Many of the downstream genes are indeed membrane proteins but does this mean that they interact with the arrest peptides? Are there signal sequences or other motifs that would be recognized by the different target proteins? It is not made clear about the different localization (cytoplasmic, membrane, etc) of the different regulatory proteins and whether they are expected to be directly involved in sensing the arrested ribosomes.

Response: As stated in our response to Reviewer 1, we consider it plausible that force-mediated arrest release plays an important role in the molecular mechanisms of how arrest peptides exert their cellular functions. Thus, we have added the following statement in Discussion (lines 506-513), as also stated in our comment to Reviewer 1.

“Many important questions regarding potential functions and mechanisms of RAPP/RGPP-containing arrest peptides remain to be addressed. For instance: How is translation arrest involved in these functions? Does arrest indeed regulate the expression of downstream genes? If so, what are the molecular mechanisms mediating this regulation? One particularly intriguing question concerns whether the force-dependent arrest release, which has been shown to occur when mechanical pulling is applied to the nascent chain (Goldman *et al*, 2015) also plays a significant role in functions beyond the monitoring of protein localization pathways. If this is the case, identifying the source of such pulling forces will be of considerable interest.”

In addition we also added the following discussion in response to the question by Reviewer 2 (line 513-519).

“In the cases of arrest peptides that monitor localization pathways, arrest release is triggered by a pulling force generated through direct interaction with the Sec or YidC machinery encoded by downstream genes. In contrast, for CruR, it has been proposed that arrest release is mediated by the interaction between CruR and copper ions (Roy *et al*, 2022). Although it is plausible that a pulling force is ultimately involved in arrest release, the mechanism by which this force is generated is not necessarily limited to direct interaction with the downstream gene product.”

Is it only the RxPP motif that is conserved in these leader peptides or are there also differential conservation of the N-terminal motifs depending on the downstream genes?

Response: To address the question raised by the reviewer, we conducted an additional bioinformatic analysis to extract characteristic features of the N-terminal regions of *S. lividans* arrest peptides. As also mentioned in our response to Reviewer 1, we identified conserved residues or motifs in the N-terminal regions of some arrest peptides (see lines 400-423 in Results, lines 494-502 in Discussion, and Appendix Fig. S6, 7). For example, arrest peptides encoded upstream of copper-related genes were found to have predicted copper-binding sites, whereas the arrest peptide encoded upstream of the gene for a thioredoxin-like protein were predicted to contain a pair of Cys residues that are proximal in the structure. These observations provide additional clues to the potential functions of the arrest peptides.

Referee #3:

The paper of Fujiwara et al. expands on the previous impressive body of work from the Chiba lab on the translation arrest peptides in bacteria. Here, the authors investigated the contribution of individual amino acids of the RAPP and RAPG arrest motifs to the translation arrest and then carried out an extensive bioinformatics analysis of the occurrence of the putative arrest motifs in the bacterial genomes. This analysis suggests that the putative arrest motifs are generally counter-selected in the bacterial proteomes. However, they are retained in short proteins whose translation may regulate expression of the downstream genes with various functions.

Critique

This is an interesting work, based on the extremely 'clean' experimental data and insightful bioinformatics analysis. I have only a couple of semi-major critics. The rest are more of a cosmetic nature.

Response: We appreciate the reviewer's positive evaluation.

Specifically, the role of the sequences N- adjacent to the stalling RAPP and RGPP motifs are not sufficiently addressed. Thus, it remains unclear, whether RGPP and possibly some other stalling tetrapeptide sequences would stall translation irrespective of the rest of the sequence context. In their previous study (ref. 20) authors partly addressed this question for the RAPP sequence. They also mention briefly in this manuscript the possible effect of the N-terminal context. However, the reader would benefit from a more explicit discussion of whether the tetrapeptide motif is sufficient (or not) for the translation arrest.

Response: As the reviewer pointed out, the role of the N-adjacent region is not yet fully understood. Although we are currently unable to make definitive statements regarding the role of the N-terminal region, we mentioned a possibility that the N-terminal interaction may contribute to arrest by reducing the fluctuation of nascent chain to stabilize the RxPP-PTC interaction. Also, in accordance with the reviewer's suggestion that "the reader would benefit from a more explicit discussion of whether the tetrapeptide motif is sufficient (or not) for the translation arrest", we have added the following statement to the Discussion section (line 455-461).

*"Our previous mutational studies have shown that the RAPP-like motif alone is insufficient to induce translation arrest, and that the adjacent N-terminal region is essential for arrest activity (Sakiyama *et al*, 2021; Morici *et al*, 2024). It is plausible that the N-terminal region contributes to arrest by interacting with the ribosome in a way that reduces nascent chain fluctuation, thereby stabilizing the interaction between the C-terminal RAPP-like motif and PTC. If such a role can be achieved through general, rather than sequence-specific interactions, it may explain the tolerance for a wide variety of sequences in the N-terminal region."*

I was also surprised why, with using a GFP-lacZ reporter, the authors did not normalize beta-galactosidase activity by GFP fluorescence. Conceivably, the mutations they introduce might affect mRNA stability; GFP-based normalization could possibly account for this.

Response: While we agree that GFP is a reasonable standard reporter, it proved

technically challenging in our *B. subtilis* system due to low expression levels. In addition, since arrest products remain tethered to the ribosome, the folding efficiency of GFP in this stalled state could possibly be inequivalent to that of the fully synthesized, free protein (see Kaiser et al., 2011, Science: doi: 10.1126/science.1209740.). For these reasons, we employed the conventional means of normalization of β -galactosidase activity by culture turbidity. Nevertheless, as the reviewer rightly pointed out, the results should be interpreted with consideration for possible effects on mRNA stability. Conversely, if GFP were to be used as a standard, interpretation would also need to take into account issues related to protein folding efficiency.

Importantly, in this study, the efficiency of translational arrest was also assessed by the in vitro system (Fig. 2F–I, Fig. 5B), allowing interpretation of the results based on both in vivo and in vitro data.

Other comments:

l. 61: either 'about' or 'nearly' not 'about nearly'

Response: Corrected.

l.99; add 'ApdA' in the parentheses as a RAPP-containing peptide.

Response: Corrected.

l. 175 replace 'know' with known'.

Response: Corrected.

l. 197. Delete 'whether'

Response: Corrected.

l. 300. Decipher 'GTDB' on the first use of the abbreviation.

Response: Corrected.

Fig. 1a and 5a. What do gray boxes within the GFP-lacZ intergenic region represent?

Response: Gray boxes represent transmembrane segments or signal sequences. We added the explanation in figure legends of Fig. 1A and 5A.

Fig. 5b. The RNase + and - signs seem to be flipped.

Response: Corrected.

Dear Dr. Chiba,

Thank you for submitting your revised manuscript for consideration by the EMBO Journal. It has now been seen by two of the three original referees whose comments are enclosed (unfortunately, referee #1 became unresponsive, so we were forced to continue without them). As you will see, both referees express interest in your manuscript, are happy with your revision and are broadly in favour of publication. Yet, there are still more technical aspects noted by our editorial assistance team that needs revision. Please find further details below.

We generally allow three months as standard revision time. Yet, I am sure you can address the remaining issues much quicker than that.

Thank you for the opportunity to consider your work for publication. I look forward to your revision.

Yours sincerely,

Yehu Moran
Academic Editor
The EMBO Journal

We realize that it is difficult to revise to a specific deadline. In the interest of protecting the conceptual advance provided by the work, we recommend a revision within 3 months (15th Dec 2025). Please discuss the revision progress ahead of this time with the editor if you require more time to complete the revisions.

Specific comments by the editorial assistance team

*DATA AVAILABILITY SECTION: included, but should be moved to the end of the Methods,

*FUNDING: please add the project number JPMJSP2157 to the last entry in the funders list in our system.

*Author Contributions: Please remove from manuscript text and keep only in the system.

* Please correct the citations in the manuscript text for the appendix tables: it should be "Appendix Table S1", etc. Please also add the figure with additional experiments to the appendix file and add a title to the figure, i.e. "Appendix Figure Sx".

*SOURCE DATA: in with completed checklist, but the main figure files should be uploaded as one (ZIP) file per figure.

*SYNOPSIS IMAGE: Currently not provided. Please provide.

*SYNOPSIS TEXT: Currently not provided. Please provide.

*FIGURE CALLOUTS: Please add callouts for all EV figure panels; there are citations for Source Data 1 and Source Data 2; should these be for Figure 1 Source Data and Figure 2 Source Data? Please correct as appropriate.

Figures: During our routine image checks, we noticed that the blot images across the figure set appear pixelated under analysis. This is a common result of converting original 16-bit TIFF images to RGB format for publication, and while not a cause for concern, it can sometimes give the impression of image alteration to critical readers.

To resolve this, please upload the figure set at a higher resolution.

Figure legends:

1. Please note that information related to n is missing in the legends of figure 4C. Please correct.
2. Please note that the error bars are not defined in the legends of figure 4C. Please correct.

Referee Reports

Referee #2:

The authors have addressed as best as possible my comments.

Referee #3:

The authors have adequately addressed the reviewers' concerns and questions, include those of me. I believe in its current version the paper is good to be published in EMBO J.

The authors addressed the remaining editorial issues.

Dear Dr. Chiba,

I am pleased to inform you that your manuscript has been accepted for publication in the EMBO Journal.

Yours sincerely,

Yehu Moran
Editor
The EMBO Journal

Please note that it is The EMBO Journal policy for the transcript of the editorial process (containing referee reports and your response letters) to be published as an online supplement to each paper. If you should prefer removal of any referee-only figures included in the point-by-point response(s), e.g. because they may still be used for future publication or because they have been reproduced from published work by others, please do let us know immediately via response email.

More information is available here: https://www.embopress.org/transparent-process#Review_Process